# Whole-brain comparison of rodent and human brains using spatial transcriptomics

Antoine Beauchamp[1,2,3]*, Yohan Yee[1,2,3], Ben C Darwin[1,2], Armin Raznahan[4], Rogier B Mars[5,6]*[†], Jason P Lerch[1,2,3,5]*[†]

[1]The Hospital for Sick Children, Toronto, Canada; [2]Mouse Imaging Centre, Toronto, Canada; [3]Department of Medical Biophysics, University of Toronto, Toronto, Canada; [4]Section on Developmental Neurogenomics, Human Genetics Branch, National Institute of Mental Health Intramural Research Program, Bethesda, United States; [5]Wellcome Centre for Integrative Neuroimaging, Centre for Functional MRI of the Brain (FMRIB), Nuffield Department of Clinical Neurosciences, John Radcliffe Hospital, University of Oxford, Oxford, United Kingdom; [6]Donders Institute for Brain, Cognition and Behaviour, Radboud University Nijmegen, Nijmegen, Netherlands

**Abstract** The ever-increasing use of mouse models in preclinical neuroscience research calls for an improvement in the methods used to translate findings between mouse and human brains. Previously, we showed that the brains of primates can be compared in a direct quantitative manner using a common reference space built from white matter tractography data (Mars et al., 2018b). Here, we extend the common space approach to evaluate the similarity of mouse and human brain regions using openly accessible brain-wide transcriptomic data sets. We show that mouse-human homologous genes capture broad patterns of neuroanatomical organization, but the resolution of cross-species correspondences can be improved using a novel supervised machine learning approach. Using this method, we demonstrate that sensorimotor subdivisions of the neocortex exhibit greater similarity between species, compared with supramodal subdivisions, and mouse isocortical regions separate into sensorimotor and supramodal clusters based on their similarity to human cortical regions. We also find that mouse and human striatal regions are strongly conserved, with the mouse caudoputamen exhibiting an equal degree of similarity to both the human caudate and putamen.

*For correspondence: antoine.beauchamp@mail. utoronto.ca (AB); rogier.mars@ndcn.ox.ac.uk (RBM); jason.lerch@ndcn.ox.ac.uk (JPL)

[†]These authors contributed equally to this work

**Competing interest:** The authors declare that no competing interests exist.

## Editor's evaluation

This important work develops new methods for aligning measures of brain-wide gene expression in the mouse and human brains. It presents compelling evidence in support of both conserved and species-specific transcriptional patterns. The work will be of interest to neuroscientists and geneticists interested in the molecular correlates of brain evolution.

## Introduction

Animal models play an indispensable role in neuroscience research, not only for understanding disease and developing treatments but also for obtaining data that cannot be obtained in the human. While numerous species have been used to model the human brain, the mouse has emerged as the most prominent of these, due to its rapid life cycle, straightforward husbandry, and amenability to genetic engineering (*Dietrich et al., 2014*; *Ellenbroek and Youn, 2016*; *Kabakci et al., 2004*; *Houdebine, 2004*). Mouse models have proven to be extremely useful for understanding diverse features of the

brain, from its molecular neurobiological properties to its large-scale network properties (*Hodge et al., 2019*; *Oh et al., 2014*; *Yao et al., 2021*). However, translating findings from the mouse to the human has not been straightforward. This is especially evident in the context of neuropsychopharmacology, where promising neuropsychiatric drugs have one of the highest failure rates in Phase III clinical trials (*Hay et al., 2014*).

Successful translation requires an understanding of how effects on the brain of the model species are likely to manifest in the brain of the actual species of interest. This is not trivial in the case of the mouse and human, as the two species diverged from a common ancestor about 80 million years ago (*Kaas, 2012*). Although common themes are apparent in the brains of all mammalian species studied to date (*Krubitzer, 2007*), there remain substantial differences between the mouse and human brain. Beyond the obvious differences in size, large parts of the human cortex potentially have no corresponding homologues in the mouse (*Preuss, 1995*). Direct comparisons across the brains of different species are further complicated by the fact that researchers from different traditions use inconsistent nomenclature to refer to similar neuroanatomical areas (*van Heukelum et al., 2020*; *Laubach et al., 2018*).

Over the course of the last decade, we have developed novel approaches to explicitly evaluate similarities and differences between the brains of related species. These approaches describe brains using common data spaces that are directly comparable between species, making it possible to evaluate the similarity of different regions in a quantitative fashion (*Mars et al., 2021*). This way, potential homologues can be formally tested, and regions of the brain that do not allow for straightforward translation can be identified (*Mars et al., 2018b*). Establishing such a formal translation between the mouse and the human brain would allow scientists involved in translational research to explicitly test hypotheses about conservation of brain regions, identify regions that are well suited to translational paradigms, and directly transform quantitative maps from the brain of one species to the other.

One approach toward building these common spaces has been to exploit connectivity. It has previously been demonstrated that brain regions can be identified via their unique set of connections to other regions in the brain. This *connectivity fingerprint* can therefore be seen as a diagnostic of an area (*Mars et al., 2018a*; *Passingham et al., 2002*). The common connectivity space approach relies on defining agreed upon neuroanatomical homologues a priori and then expressing the connectivity fingerprint of regions under investigation with those established homologues in the two brains (*Mars et al., 2016a*). The connections of any given region to the established homologues thus form a common space, which links the two brains. In a series of early studies, we compared the connectivity of the macaque and human brain, identifying homologies as well as specializations across association cortex (*Mars et al., 2013*; *Neubert et al., 2014*; *Sallet et al., 2013*). The same approach has recently been applied to mouse-human comparisons for the first time, demonstrating conserved organization between the mouse and human striatum, but some specialization in the human caudate related to connectivity with the prefrontal cortex (*Balsters et al., 2020*). A similar study recently compared connectivity of the medial frontal cortex across rats, marmosets, and humans (*Schaeffer et al., 2020*). However, the lack of established neuroanatomical homologues in mice, particularly in the cortex, limits the use of connectivity to compare these species.

A more promising approach to mouse-human comparisons could be to exploit the spatial patterns of gene expression. Advances in transcriptomic mapping can be used to characterise the differential expression of many thousands of genes across the brain and compare the pattern between regions (*Ortiz et al., 2020*). Moreover, the availability of whole-brain spatial transcriptomic data sets for multiple species provides an opportunity to run novel analyses at low cost (*Hawrylycz et al., 2012*; *Lein et al., 2007*). Such maps for the human cortex show topographic patterns that mimic those observed in other modalities, such as a gradient between primary and heteromodal areas of the neocortex (*Burt et al., 2018*). Importantly, these patterns appear to be conserved across mammalian species (*Fulcher et al., 2019*), which opens up the possibility of using the expression of homologous genes as a common space across species. In fact, a recent study demonstrated how the expression of homologous genes can be used to directly register mouse and vole brains into a common reference frame, which allows for direct point-by-point comparisons of brain maps (*Englund et al., 2021*). However, this specific approach is only feasible because of the large degree of morphological similarity between mouse and vole brains. In the case of mouse-human comparisons, we almost certainly

cannot directly register mouse and human brains into a common coordinate frame using methods for image registration. Hence we need to be more creative in our approach.

Here we examine the patterns of similarity between the mouse and human brain using a common space constructed from spatial gene expression data sets. We begin with an initial set of 2835 homologous genes. Subsequently, we present and evaluate a novel method for improving the resolution of mouse-human neuroanatomical correspondences using a supervised machine learning approach. Using the novel representation of the gene expression common space, that is, a latent gene expression space, we analyze the similarity of mouse and human isocortical subdivisions and demonstrate that sensorimotor regions exhibit a higher degree of similarity than supramodal regions. Finally, we examine the patterns of transcriptomic similarity at a voxel-wise level in the mouse and human striatum.

## Results
### Homologous genes capture broad similarities in the mouse and human brains

We first examined the pattern of similarities that emerged when comparing mouse and human brain regions on the basis of their gene expression profiles. We constructed a gene expression common space using widely available data sets from the Allen Institute for Brain Science: the Allen Mouse Brain Atlas (AMBA) and the Allen Human Brain Atlas (AHBA) (*Hawrylycz et al., 2012*; *Lein et al., 2007*). These data sets provide whole-brain coverage of expression intensity for thousands of genes in the mouse and human genomes. For our purposes, we filtered these gene sets to retain only mouse-human homologous genes using a list of orthologues obtained from the NCBI HomoloGene system (*NCBI Resource Coordinators, 2018*). Using a gene enrichment analysis, we found that this reduced gene set was significantly associated with a number of biological processes related to the nervous system, with Gene Ontology labels such as 'nervous system development', 'neurogenesis', and 'regulation of nervous system development'. Additional modules returned with high significance were 'regulation of multicellular organismal process', 'regulation of biological quality', and 'multicellular organism development'. The full set of significant modules can be found in *Supplementary file 1*.

Prior to analysis, the mouse and human homologous gene expression data sets were pre-processed using a pipeline that included quality control checks, normalization procedures, and aggregation of the expression values under a set of atlas labels. The result was a gene-by-region matrix in either species, describing the normalized expression of 2835 homologous genes across 67 mouse regions and 88 human regions (see Materials and methods). We quantified the degree of similarity between all pairs of mouse and human regions using the Pearson correlation coefficient, resulting in a mouse-human similarity matrix (*Figure 1A*).

We find that the similarity matrix exhibits broad patterns of positive correlation between the mouse and human brains. These clusters of similarity correspond to coarse neuroanatomical regions that are generally well defined in both species. For instance, we observe that, overall, the mouse isocortex is similar to the human cerebral cortex, with the exception of the hippocampal formation, which forms a unique cluster. Similarly, the mouse and human cerebellar hemispheres cluster together, while the cerebellar nuclei show relatively high correlation to each other ($r=0.351$) as well as to brain stem structures like the pons ($r = 0.328$ and $r = 0.335$ for the mouse and human nuclei, respectively) and myelencephalon ($r = 0.288$ and $r = 0.351$). The associations between broad regions such as these are self-evident in the correlation matrix.

Our ability to resolve regional matches on a finer scale is limited when using all homologous genes in this way. This is especially true for regions within the cerebral and cerebellar cortices, which exhibit a high degree of internal homogeneity. This is apparent in the similarity profiles, defined here as the set of correlation values between a given seed region and all target regions in the other species. For example, the human precentral gyrus and cuneus are most strongly correlated to many regions of the mouse isocortex. While the brain maps feature a rostral-caudal gradient (*Figure 1B*), the profiles of the two seeds are highly similar despite the regions having very different functions (*Figure 1C*). Indeed, the correlation between the similarity profiles of the precentral gyrus and cuneus is $r = 0.975$. The similarity profile of human cerebellar crus 1 highlights another example of this homogeneity. The profile of crus 1 is similar to that of all regions of the mouse cerebellum, with an average correlation of $r = 0.213$ and a standard deviation of $\sigma = 0.034$. Across all regions, the variance of the correlations

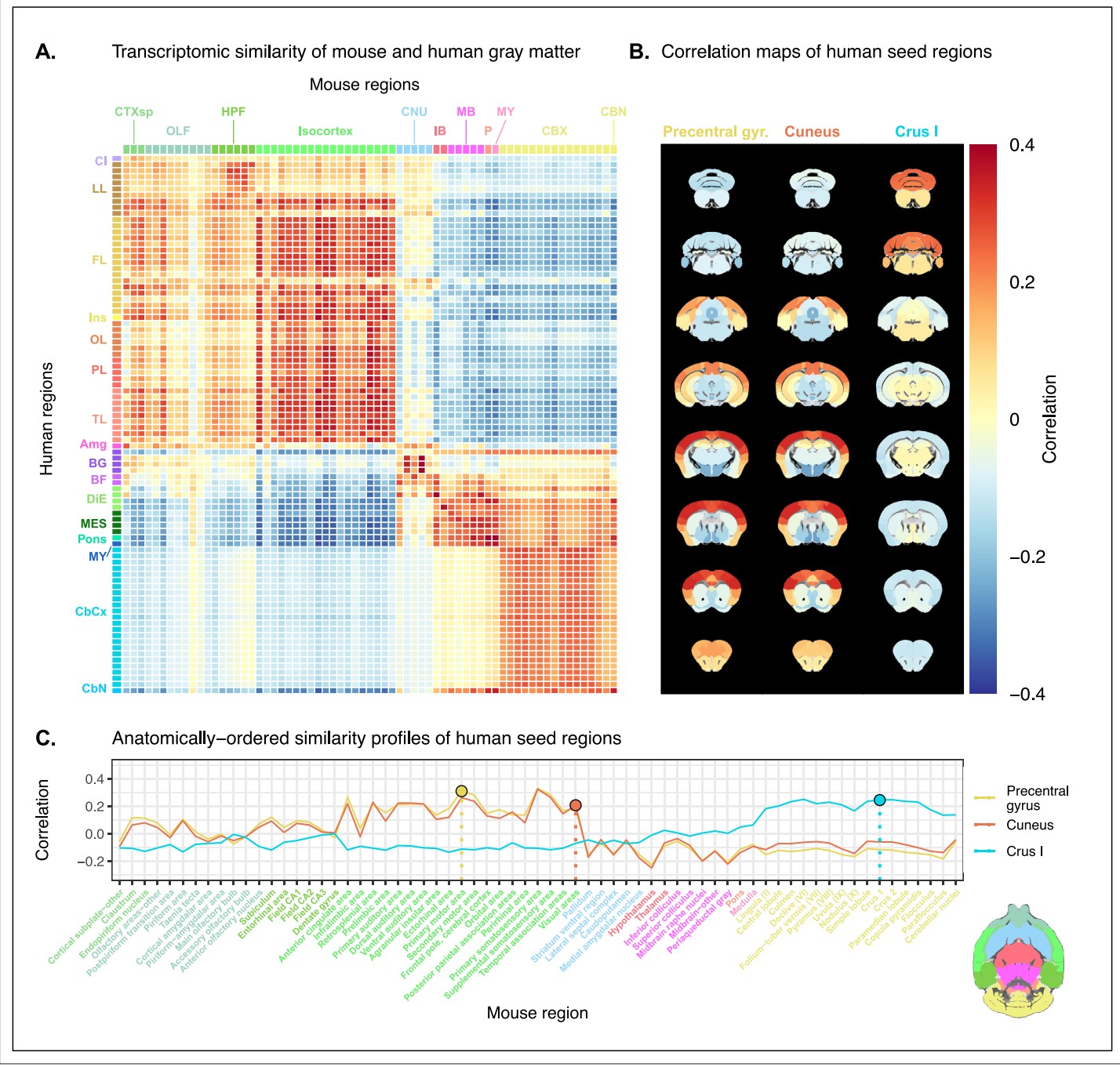

**Figure 1.** Transcriptomic similarity in the mouse and human brains. (**A**) Similarity matrix displaying the correlation between 67 mouse regions and 88 human regions based on the expression of 2835 homologous genes. Columns are annotated with 11 broad mouse regions: cortical subplate (CTXsp), olfactory areas (OLF), hippocampal formation (HPF), isocortex, cerebral nuclei (CNU), interbrain (IB), midbrain (MB), pons (**P**), medulla (MY), cerebellar cortex (CBX), and cerebellar nuclei (CBN). Rows are annotated with 16 broad human regions: claustrum (Cl), limbic lobe (LL), frontal lobe (FL), insula (Ins), occipital lobe (OL), parietal lobe (PL), temporal lobe (TL), amygdala (Amg), basal ganglia (BG), basal forebrain (BF), diencephalon (DIE), mesencephalon (MES), pons, myelencephalon (MY), cerebellar cortex (CbCx), and cerebellar nuclei (CbN). Broad patterns of similarity are evident between coarsely defined brain regions, while correlation patterns are mostly homogeneous within these regions. (**B**) Mouse brain coronal slices showing similarity profiles for the human precentral gyrus, cuneus, and crus I. Correlation patterns for the precentral gyrus and cuneus are highly similar to one another and broadly similar to most isocortical regions. The crus I is homogeneously similar to the mouse cerebellum. (**C**) Anatomically ordered line charts displaying the similarity profiles for the seed regions in (**B**). Dashed vertical lines indicate the canonical mouse homologue for each human seed. Annotation colors correspond to atlas colors from the Allen Mouse Brain Atlas and Allen Human Brain Atlas for mouse and human regions, respectively.

The online version of this article includes the following source data for figure 1:

**Source data 1.** Mouse–human similarity matrix using homologous genes.

across cortical regions is $\sigma^2 = 0.0067$ while that across cerebellar hemispheric regions is $\sigma^2 = 0.0013$, compared with a total variation of $\sigma^2 = 0.031$ across all entries in the matrix.

Although there is distinguishing power in the profiles of regions at a finer scale, this is much smaller than between coarse anatomical regions. This is also true for parts of the broad anatomical systems that are part of the same functional system. This suggests that the regional expression patterns of mouse-human homologous genes can be used to identify general similarities between the brains of the two species using a simple correlation measure, but the ability to identify finer scale matches might require a more subtle approach.

## A latent gene expression space improves the resolution of mouse-human associations

In the previous analyses, we showed that the expression profiles of homologous genes capture broad similarities across the mouse and the human for the major subdivisions of the brain. Some information at a finer resolution (e.g. within the isocortex) was also evident but much less distinctive. Our next goal was to investigate whether it is possible to leverage the gene expression data sets to relate mouse and human brains to one another at a finer regional level. In order to do so, we sought to maximize the informational value in the set of 2835 homologous genes by creating a new latent common space that exploits the regional distinctiveness of the expression profiles.

The approach used in the previous analysis relied on using homologous genes as a common space between the mouse and human brain. This approach effectively assigns equal value to each gene, whereas a more powerful approach would be to weight genes by their ability to distinguish between different brain regions. We investigated whether we could accomplish this by constructing a new set of variables from combinations of the homologous genes. Our primary goal here was to transform the initial gene space into a new common space that would improve the locality of the matches. However, while we sought a transformation that would allow us to recapitulate known mouse-human neuroanatomical homologues, we also wanted to avoid directly encoding such correspondences in the transformation. Using this information as part of the optimization process for the transformation would run the risk of driving the transformation toward mouse-human pairs that are already known. While we are interested in being able to recover such matches, we are equally interested in identifying novel and unexpected associations between neuroanatomical regions in the mouse and human brains (e.g. one-to-many correspondences). Given these criteria, our approach to identifying an appropriate transformation was to train a multi-layer perceptron classifier on the data from the AMBA. The classifier was tasked with predicting the 67 labels in our mouse atlas from the voxel-wise expression of the homologous genes (*Figure 2A*).

While the model could have been trained using the data from either species, we chose to use the mouse data because it provides continuous coverage of the entire brain and is thus better suited to this purpose. In training the model to perform this classification task, we effectively optimize the network architecture to identify a transformation from the input gene space to a space that encodes information about the delineation between mouse brain regions. To extract this transformation, we removed the output layer from the trained neural network. The resulting architecture defines a transformation from the input space to a lower-dimensional gene expression latent space. We then applied this transformation to the mouse and human gene-by-region expression matrices to obtain representations of the data in the latent common space (*Figure 2B*). Finally, we used these gene expression latent common space matrices to compute the new similarity matrix (*Figure 2C*). Since the optimization algorithm used to train the perceptron features an inherent degree of stochasticity, we repeated this training and transformation process 500 times to generate a distribution of latent spaces and similarity matrices over training runs. Although the neural network and associated latent space do not directly provide information about which genes are most important for the classification of specific mouse atlas labels, this type of information can be derived from the model using attribution methods such as integrated gradients (*Figure 2—figure supplement 1*; *Sundararajan et al., 2017*). Each brain region in the classification task is associated with the input genes in different ways, such that there isn't a single weighting of gene importance for the entire model. While most genes contribute to the classification of any given label in some capacity, it is often the case that the network relies on a reduced subset of genes to arrive at a decision. For example, the genes, *Prrg2* and *Cd4,* were found to be the most influential for the classification of the caudoputamen, when the feature attributions were

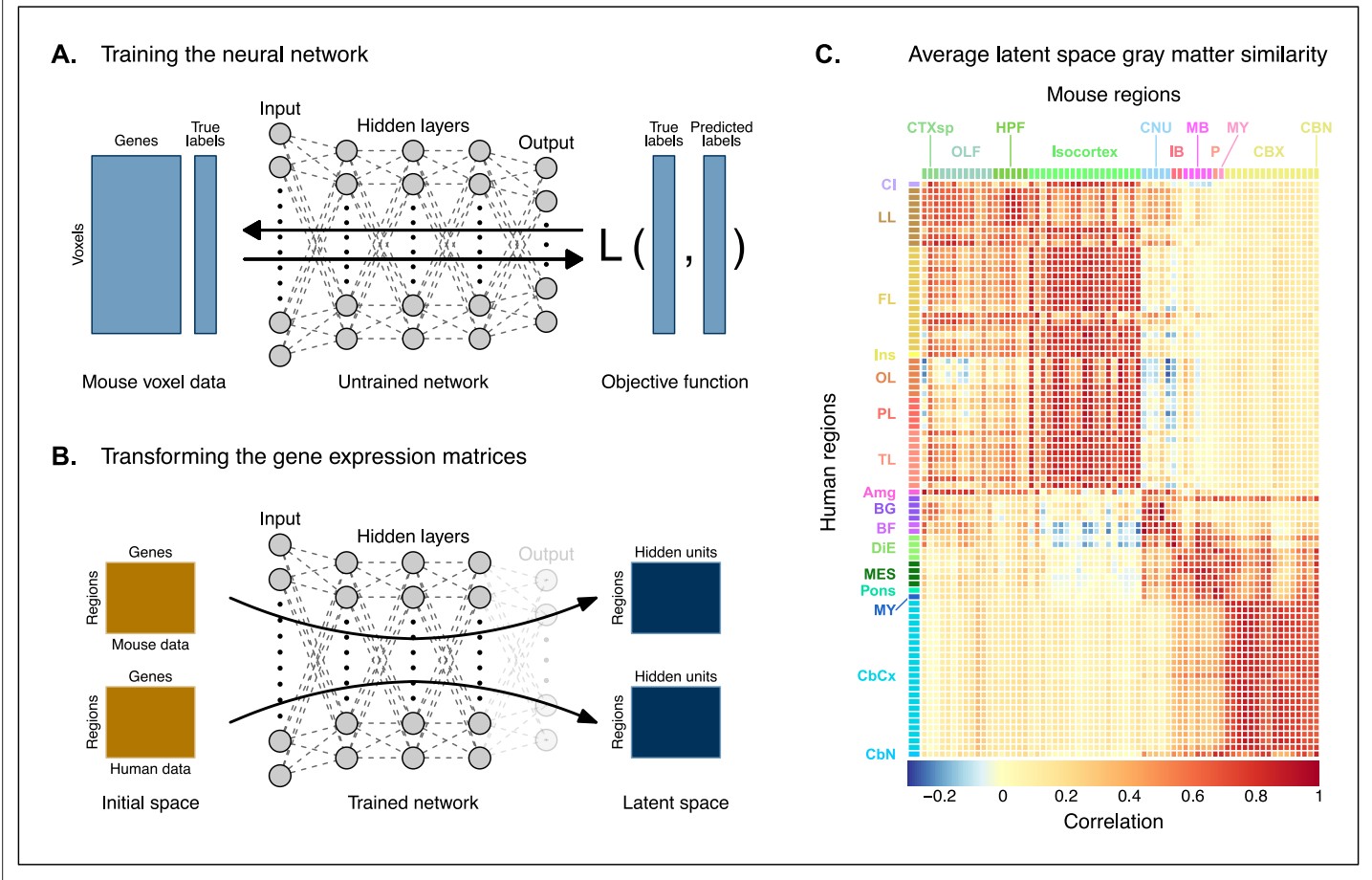

**Figure 2.** Creating a new common space. (**A**) Voxel-wise expression maps from 2835 homologous genes in the Allen Mouse Brain Atlas were used to train the neural network to classify each mouse voxel into one of the 67 atlas regions. (**B**) Once the network is trained, the output layer is removed. The mouse and human regional gene expression matrices are passed through the network, resulting in lower-dimensional latent space representations of the data. The training and transformation process were repeated 500 times. (**C**) A similarity matrix displaying the gene expression latent space correlation between mouse and human regions, averaged over 500 neural network training runs. Similar brain regions exhibit very high correlation values. Column and row annotations as described in *Figure 1*.

The online version of this article includes the following source data and figure supplement(s) for figure 2:

**Source data 1.** Correlations between mouse and human brain regions in all latent spaces (1 of 3), related to *Figure 2C*.

**Source data 2.** Correlations between mouse and human brain regions in all latent spaces (2 of 3), related to *Figure 2C*.

**Source data 3.** Correlations between mouse and human brain regions in all latent spaces (3 of 3), related to *Figure 2C*.

**Figure supplement 1.** Multi-layer perceptron feature importance for the classification of the caudoputamen (**A**), the primary motor area (**B**), and the infralimbic area (**C**).

averaged over all training runs. In contrast, *Rfx4* and *Glra3* were the most influential for the classification of the primary motor area. In some cases, the spatial expression pattern of the gene clearly shows a demarcation of the region of interest (e.g. *Cd4*), but this is not always the case, nor is it necessary, as the network learns from the entire gene expression signature of all voxels.

To assess whether the latent space representations of the data improved the resolution of the mouse-human matches, we considered two criteria. The first was whether the similarity profiles of the mouse atlas regions were more localized within the corresponding broad regions of interest (e.g. primary motor area within isocortex), compared with their similarity profiles in the original gene space. We term this the locality criterion. The second criterion was whether the degree of similarity between canonical neuroanatomical homologues improved in this new latent common space. We term this as the homology criterion. The locality criterion tells us about our ability to extract finer-scale signal in these profiles, while the homology criterion informs us about our ability to recover expected matches

in this finer-scale signal. To evaluate these criteria, we computed ranked similarity profiles for every region in the mouse brain, ordered such that a rank of 1 indicates the most similar human region. In addition, given the difference in absolute value between the input gene space and gene expression latent space correlations, we scaled the similarity profiles to the interval $[0, 1]$ in order to make comparisons between the spaces.

We evaluated the locality criterion by examining the decay rate of the top of the similarity profiles. We reasoned that the plateau of similarity to a broad brain region, as seen in the anatomically ordered similarity matrices and profiles (*Figure 1A, C*; *Figure 2C*), would correspond to a similar plateau at the head of the rank-ordered profiles. Moreover, the emergence of local signal would manifest as an increase in the range between the peaks and troughs within the broad region. In the rank-ordered profiles, this would correspond to a faster rate of decay at the head of the profile. In order to quantify this decay, we computed the rank at which each region's similarity profile decreased to a scaled value of 0.75. This was calculated for every mouse region in the initial gene space, as well as in each of the 500 gene expression latent spaces. As a measurement of performance between the two representations of the data, we then took the difference in this rank between each of the latent spaces and the original gene space (*Figure 3A*). A negative rank difference indicates an improvement in the latent space.

Examining the structure-wise distributions of these rank differences, we found that for the majority of regions in our mouse atlas, the classification approach resulted in either an improvement in the amount of locality within a broad region, or no difference from the original gene space (*Figure 3B, C*). We quantified the improvement overall by fitting a logistic regression model with no predictors to the mean rank differences of each of the atlas regions. We considered the success condition for the Bernoulli trials to be a mean rank difference less than or equal to zero. The model estimate for the Bernoulli probability – which we denote $p_B$ to distinguish from the p-value p – was $p_B = 0.78$ with a 95% CI of $[0.66, 0.86]$ . In other words, 52 of the 67 brain regions saw an improvement on average when using the latent spaces. The probability of obtaining at least as many successes as this under the null model, i.e. a binomial distribution with $p_B = 0.50$ and $n = 67$, is $p=8.64 \cdot 10^{-7}$ . We additionally evaluated the same kind of logistic regression on a region-wise basis to quantify how often the latent spaces resulted in an improvement for individual brain regions (*Figure 3C*). We found that for 46 regions (69%), the model estimated the probability to be at least at high as $p_B = 0.95$. While confidence intervals varied around this estimate, the range between the upper and lower bound was only ever as high as 0.04. For 53 of the 67 regions (79%), the q-values, i.e. p-values adjusted for multiple comparisons, were effectively null, with the largest being $q = 3.77 \cdot 10^{-16}$ . Of the remaining 14 regions, 13 had q-values equal to 1 and one region, the periacqueductal gray, had a q-value of $q = 0.854$. The regions with the smallest estimates for the Bernouilli probabilities are the dentate gyrus ($p_B = 0.0$, no variance, $q = 1$), the striatum ventral region ($p_B = 0.016$, 95% CI $[0.008, 0.032]$ , $q = 1$), and the lateral septal complex ($p = 0.016$, 95% CI $[0.008, 0.032]$ , $q = 1$). The remaining regions with $q = 1$ are all subcortical and fall under the broad subdivisions of cerebral nuclei, olfactory areas, interbrain, midbrain, pons, medulla, and cerebellar nuclei. Beyond this binary measure of improvement, some regions exhibited a large range of differences in rank over the various latent spaces. In particular regions like the main olfactory bulb (mean rank difference of $\mu = 10$, 95% CI $[-12, 33]$) and accessory olfactory bulb $\mu = 9$, 95% CI $[-13, 31]$ exhibit a substantial degree of variance. Other than these two areas, regions within the olfactory areas (e.g. piriform area) were among those that benefited the most from the classification approach, showing improvement in all sampled latent spaces, with all Bernouilli probability estimates equal to 1 and all q-values equal to 0. While the effects, i.e. rank differences, are smaller, the similarity profiles of regions belonging to the isocortex and cerebellar cortex also saw an improvement in locality. All models for isocortical areas returned Bernouilli probability estimates greater than $p_B = 0.85$ and q-values that were at most $q = 1.35 \cdot 10^{-67}$. Moreover, 9 of the 19 isocortical regions were improved in all latent spaces, that is, $p_B = 1$. Brain regions belonging to the cerebellar cortex saw similar improvement. In contrast, regions belonging to the cerebral nuclei, the diencephalon, midbrain, and hindbrain did not see much improvement in this new common space, with an average Bernouilli probability estimate of $p_B = 0.36$ for this subset. Other than the caudoputamen ($p_B = 0.99$, 95% CI $[0.97, 1.00]$ , $q = 1.35 \cdot 10^{-139}$), the superior colliculus ($p_B = 0.90$, 95% CI $[0.87, 0.92]$ , $q = 9.82 \cdot 10^{-81}$), and the inferior colliculus ($p_B = 0.75$, 95% CI $[0.71, 0.78]$, $q = 3.12 \cdot 10^{-30}$), all regions in this subset return q-values equal to 1. For many such regions, the degree of locality appears to

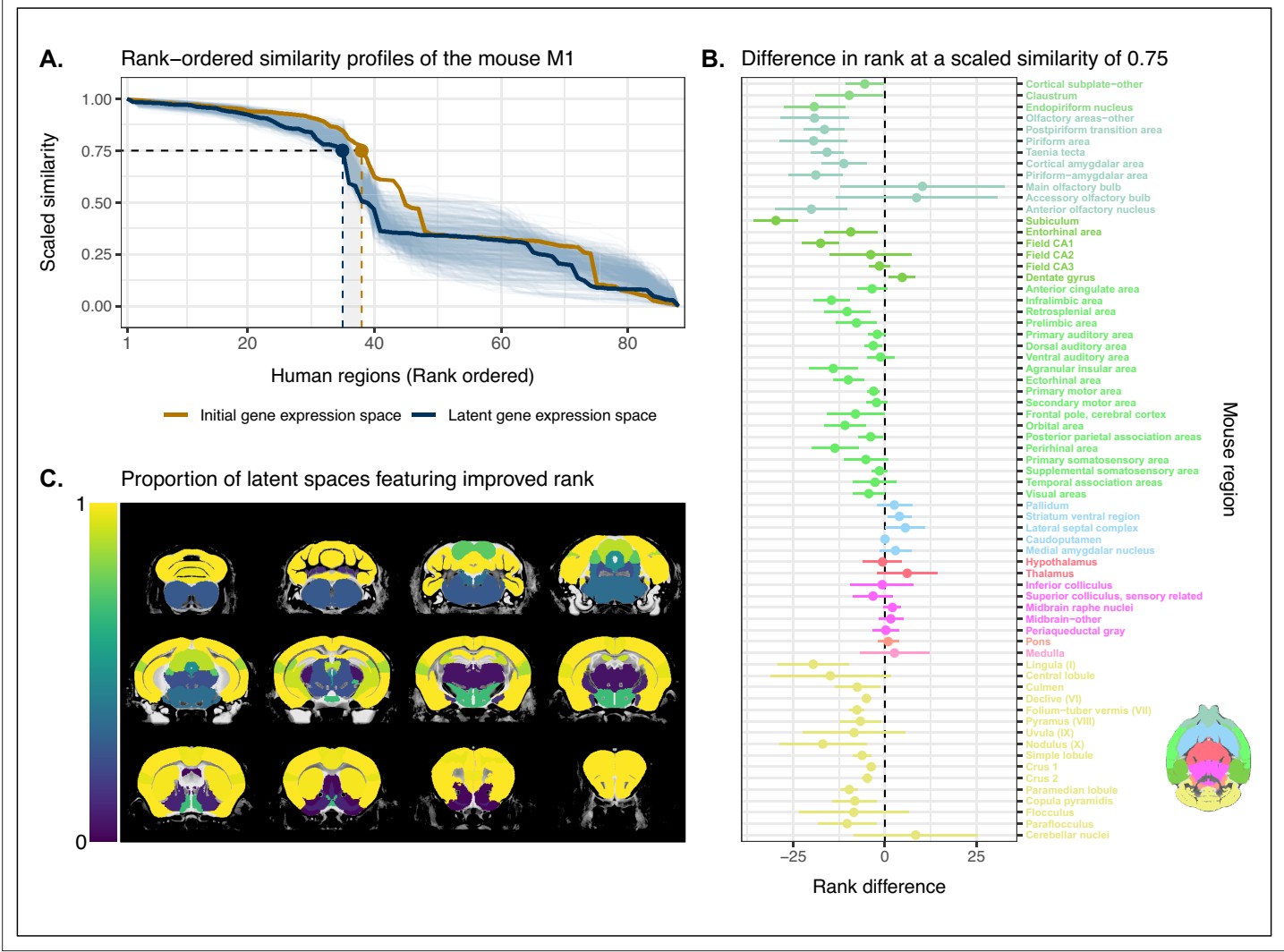

**Figure 3.** Quantifying improvement in locality in gene expression latent space. (**A**) The amount of local signal within a broadly similar region of the brain for a finer seed region's (e.g. primary motor area) similarity profile can be quantified by the decay rate of the head of the rank-ordered profile. Decay rate was quantified by computing the rank at a similarity of 0.75. This metric was compared between the initial gene expression space (orange line) and every gene expression latent space resulting from repeated training of the neural network (every blue line is a training outcome, heavy blue line serves as an example). A negative difference between these rank metrics indicates an improvement in locality in the latent space. (**B**) Structure-wise distributions of differences in rank at a similarity of 0.75 between the initial gene expression space and the gene expression latent spaces. Points and error bars represent mean and 95% CI with n = 500. Dashed black line at 0 indicates the threshold for improvement in one space over the other. Colors correspond to Allen Mouse Brain Atlas annotations as in *Figures 1 and 2*. Binomial likelihood (logistic regression) estimate of $p_B = 0.78$ with 95% CI [0.66, 0.86]. The probability of obtaining at least these many successes under the null binomial distribution, $B(67, 0.5)$, is p=8.64 · 10⁻⁷ . (**C**) Proportion of perceptron training runs resulting in an improvement or null difference in the gene expression latent space compared with the initial space, estimated using region-wise logistic regressions. Cortical and cerebellar regions exhibit high proportions of improvement, while subcortical regions are less likely to be improved by the classification process.

The online version of this article includes the following source data for figure 3:

**Source data 1.** Scaled similarity profiles of the mouse primary motor area, related to *Figure 3A*.

**Source data 2.** Ranks at a similarity of 0.75 for mouse regions in the homologous gene space and all latent spaces, related to *Figure 3B*.

**Source data 3.** Logistic regression model estimates for mouse regions, related to *Figure 3C*.

be worse in this space, though only by a small number of ranks, for example, striatum ventral region (mean rank difference of $\mu = 4$, 95% CI $[1, 7]$) and lateral septal complex ($\mu = 6$, 95% CI $[0, 11]$). Indeed, computing the average rank difference over this subset of regions across all latent spaces, we find

$\mu = 2$ with 95% CI $[-5, 8]$. These results demonstrate that the supervised learning approach used here can improve the resolution of neuroanatomical correspondences between the mouse and human brains, though the amount of improvement varies over the brain. Regions that were already well characterized using the initial set of homologous genes (e.g. subcortical regions) did not benefit tremendously, but numerous regions in the cortical plate and subplate, as well as the cerebellum, saw an improvement in locality in this new common space.

While the supervised learning approach improved our ability to identify matches on a finer scale for a number of brain regions, this does not necessarily mean that those improved matches are biologically meaningful. The second criterion for evaluating the performance of the neural network addresses whether this improvement in locality captures what we would expect in terms of known mouse-human homologies. To this end, we examined the degree of similarity between established mouse-human neuroanatomical pairs, both in the initial gene expression space and in the set of latent spaces. We began by establishing a list of 36 canonical mouse-human homologous pairs on the basis of common neuroanatomical labels in our atlases. For each of these regions in the mouse brain, we compared the rank of the canonical human match in the rank-ordered similarity profiles between the latent spaces and the original gene expression space (*Figure 4A*). The lower the rank, the more similar the canonical pair, with a rank of 1 indicating maximal similarity. As described above, we evaluated the overall performance of the classification approach by running a logistic regression using the average latent space rank difference over all regions in our subset. Here we find an estimated Bernouilli probability of $p_B = 0.64$ with 95% CI $[0.47, 0.78]$. Under the null binomial distribution, $B(36, 0.5)$, the probability of getting at least as many successes as this is p=0.033. We also evaluated the model for each brain region and found that 30 of the 36 regions (83%) return Bernouilli probability estimates of at least $p_B = 0.80$. Under the null binomial distribution, $B(500, 0.5)$, we find that the largest q-value among these 30 regions is $q = 4.39 \cdot 10^{-54}$. Moreover, 24 regions (67%) return Bernouilli probability estimates of at least $p_B = 0.90$, and 8 regions show improvement in all latent spaces, that is, $p_B = 1$ and $q = 0$ (*Figure 4B*). Among these 8 regions are the claustrum, the piriform area, the primary motor and somatosensory areas, and the crus 2. Additional examples of the many regions that demonstrate improvement include: the primary auditory area ($p_B = 0.83$, 95% CI $[0.80, 0.86]$, $q = 1.80 \cdot 10^{-55}$), the pallidum ($p_B = 0.86$, 95% CI $[0.83, 0.89]$, $q = 3.63 \cdot 10^{-65}$), and the crus 1 ($p_B = 0.92$, 95% CI $[0.90, 0.94]$, $q = 7.68 \cdot 10^{-95}$). Once again we find that many regions in the sub-cortex do not benefit greatly from the gene expression latent spaces, since the initial gene set was already recapitulating the appropriate match with maximal similarity. We find that the striatum ventral region, caudoputamen, hypothalamus, and pons are maximally similar to their canonical matches in at least 95% of latent spaces. In such cases, the classification approach performs as well as the original approach. While these probability estimates provide a sense of how often an improvement is returned, it is important to note that many regions in this set exhibit a substantial degree of variance over the latent spaces in the ranking of the canonical pairs, for example, the primary auditory area ($\mu = 9$, 95% CI $[1, 19]$), the visual areas ($\mu = 18$, 95% CI $[7, 29]$), and the paraflocculus ($\mu = 16$, 95% CI $[2, 29]$). This is especially apparent for cerebellar regions, indicating some instability in the neural network's ability to recover these matches.

Together, these results demonstrate that the multi-layer perceptron classification approach improves our ability to resolve finer scale mouse-human neuroanatomical matches within the broadly similar regions obtained using the initial gene expression space. By training a classifier to predict the atlas labels in one species, we were able to generate a new common space that amplified the amount of local signal within broadly similar regions while also improving our ability to recover known mouse-human neuroanatomical pairs.

## Cortical areas involved in sensorimotor processing show greater transcriptomic similarity than supramodal areas

It is well established that the brains of most, if not all, extant mammalian species follow a common organizational blueprint inherited from an early mammalian ancestor (*Kaas, 2011a*). A number of cortical subdivisions have consistently been identified in members of many distantly related mammalian species (*Krubitzer, 2007*) and hypothesized to have been present in the common ancestor of all mammals (*Kaas, 2011a*). While it is clear that basic sensorimotor cortical regions are found in the majority of mammals, including mice and humans, there is much debate about the extent to which cortical areas involved in supramodal processing are conserved across mammalian taxa. Although

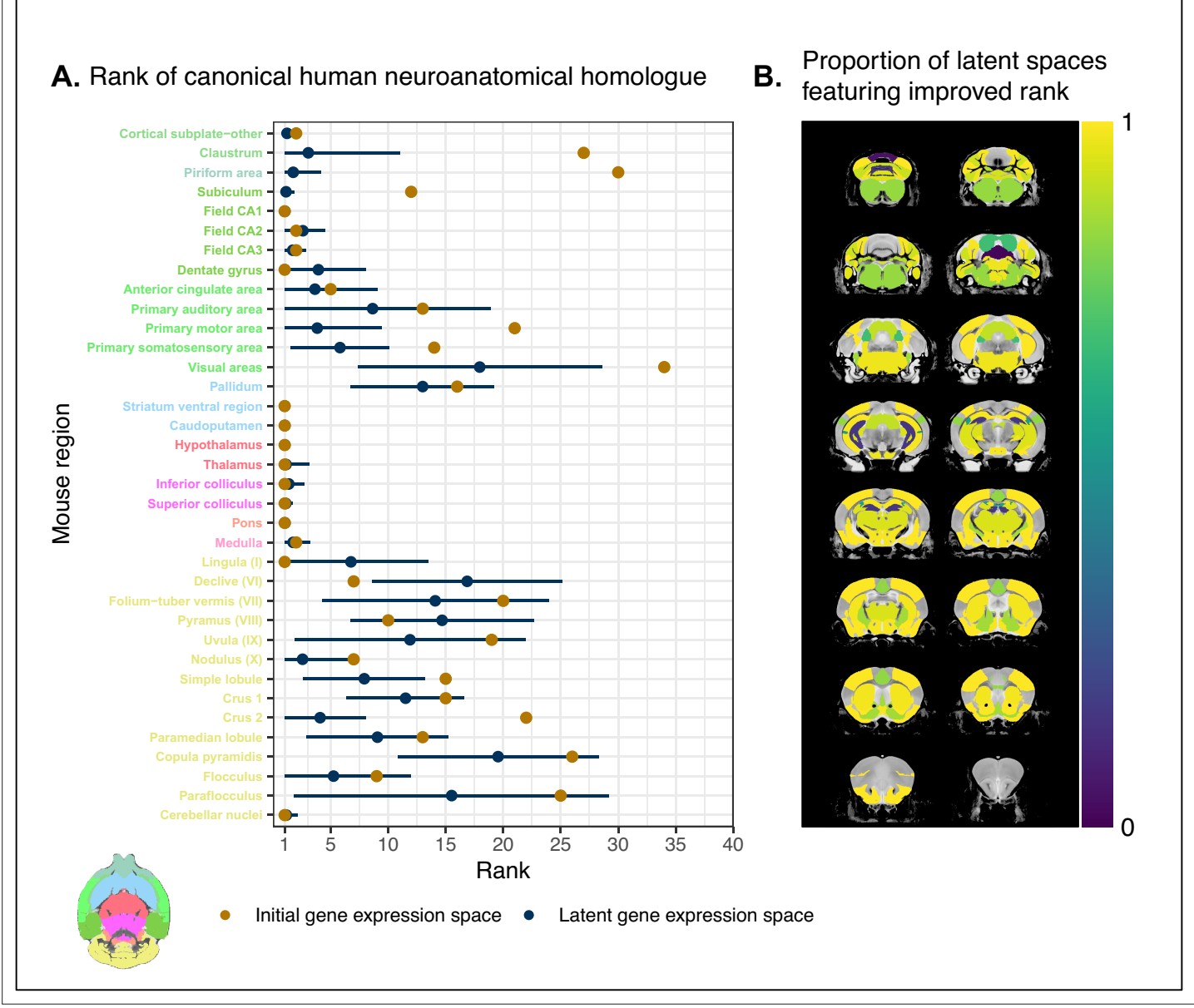

**Figure 4.** Recovering canonical neuroanatomical pairs in gene expression space. (**A**) Comparison between the ranks of canonical human matches for mouse seed regions between the initial gene expression space and gene expression latent spaces. Points and error bars represent mean and 95% CI with n = 500. Mouse region names are colored according to the Allen Mouse Brain Atlas palette. Binomial likelihood estimate of p=0.64 with 95% CI [0.47, 0.78]. The probability of obtaining at least thse many successes under the null binomial distribution, $B\left(36, 0.5\right)$, is p=0.033. (**B**) Proportion of latent spaces resulting in an improvement or null difference compared with the initial gene space, estimated using region-wise logistic regressions. Uncolored voxels correspond to regions with no established canonical human match.

The online version of this article includes the following source data for figure 4:

**Source data 1.** Ranks of canonical neuroanatomical pairs for mouse regions in the homologous gene space and all latent spaces, related to *Figure 4A*.

**Source data 2.** Logistic regression model estimates for mouse regions, related to *Figure 4B*.

some supramodal regions were likely present in the earliest mammals, including some cingulate regions and an orbitofrontal cortex (*Kaas, 2011a*), since the divergence of mouse and human lineages some 80 million years ago, the primate neocortex has undergone substantial expansion and re-organization (*Kaas, 2012*). Indeed, when comparing the human neocortex even to primate model species, this is the likely locus of areas that cannot be easily translated between species (*Mars et al., 2018b*).

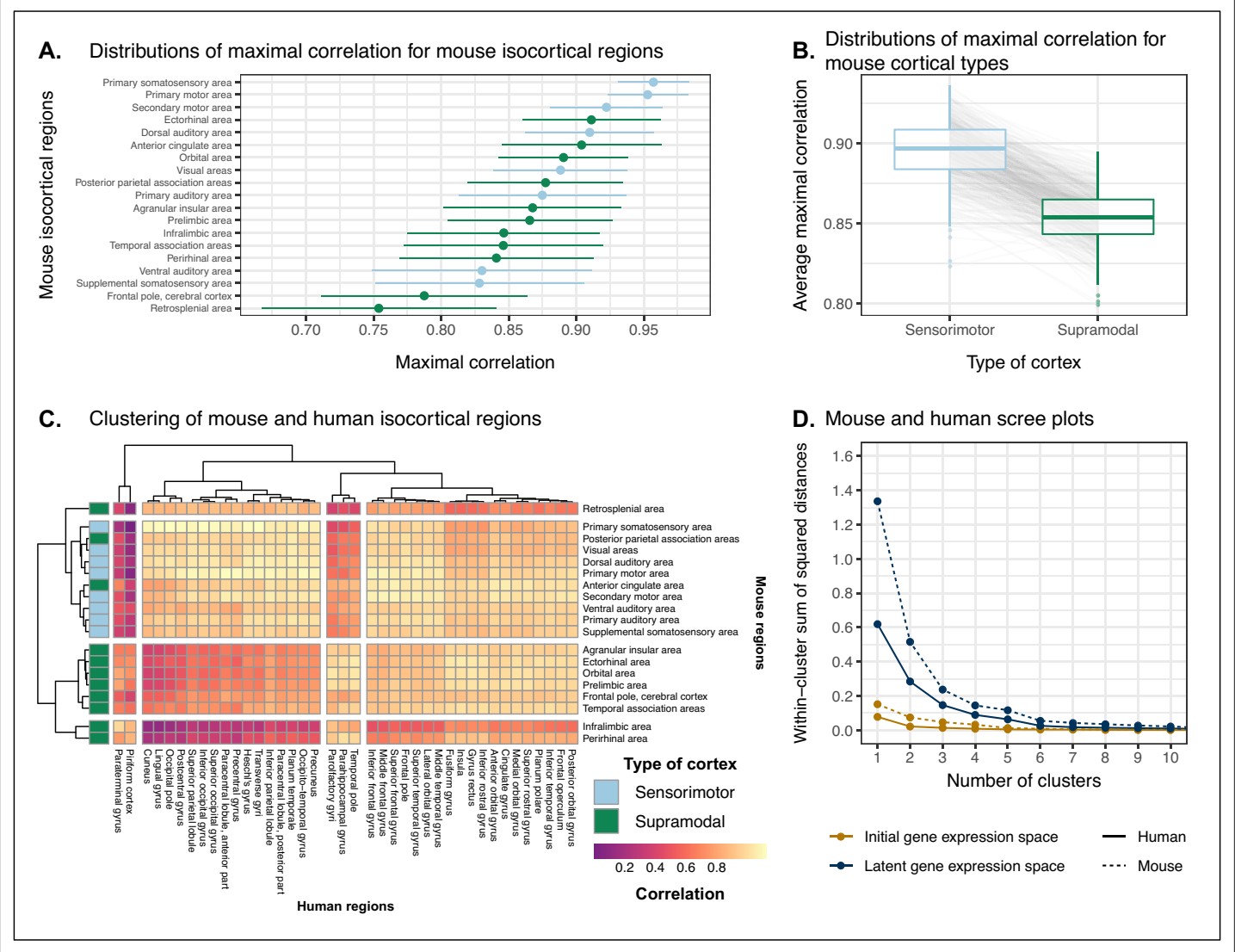

**Figure 5.** Similarity of mouse-human isocortical regions. (**A**) Maximal correlation distributions of mouse isocortical regions. Points and error bars represent mean and 95% CI over n = 500 latent space samples. Linear regression using average maximal correlation values: $\beta = -0.042$, 95% CI $[-0.087, 0.003]$, $t(17) = -1.854$, p=0.0812. (**B**) Distributions of average maximal correlation for sensorimotor and supramodal isocortical areas in each gene expression latent space. Gray lines correspond to individual latent spaces. Linear mixed-effects regression: $\beta = -0.042$, 95% CI $[-0.044, -0.040]$, $t(499) = -49.9$, p<2 · 10$^{-16}$. (**C**) Hierarchical clustering of mouse and human isocortical regions based on average latent space correlation values. Mouse regions are annotated as sensorimotor or supramodal. Four clusters were chosen for visualization using the elbow method. (**D**) Within-cluster sum of squared distances for different numbers of mouse and human isocortical clusters in the average latent space and initial homologous gene space.

The online version of this article includes the following source data and figure supplement(s) for figure 5:

**Source data 1.** Maximal correlations of mouse isocortical regions in all latent spaces, related to *Figure 5A and B*.

**Source data 2.** Correlations between mouse and human isocortical regions in all latent spaces, related to *Figure 5C*.

**Source data 3.** Scree plot data, related to *Figure 5D*.

**Figure supplement 1.** Comparison between the ranks of canonical human matches for mouse cortical seed regions in various gene expression spaces.

**Figure supplement 2.** Similarity of mouse-human isocortical regions in latent spaces obtained using only cortical labels.

As a result, it is important to investigate whether our between-species mapping is more successful in somatosensory areas than supramodal areas.

We assessed the similarity between mouse and human isocortical areas using the pairwise correlations in each of the gene expression latent spaces returned from the multi-layer perceptron. For every region in the mouse isocortex, we evaluated the distribution of maximal correlation values over latent

spaces (*Figure 5A*). While the region-wise variance for each isocortical area was large, we found that, on average, sensorimotor regions exhibited higher maximal correlation values than supramodal regions (linear regression with binary predictor: $\beta = -0.042$, 95% CI $[-0.087, 0.003]$, $t(17) = -1.854$, p=0.0812). The mouse primary somatosensory ($r = 0.96$, 95% CI $[0.93, 0.98]$) and motor ($r = 0.95$ with 95% CI $[0.92, 0.98]$) areas have the highest average maximal correlation values. We additionally examined the distributions of maximal correlation, grouped by cortex type (*Figure 5B*). To generate these distributions, we computed average maximal correlation values by cortex type in each of the latent spaces. Here too we find that sensorimotor regions are associated with higher maximal correlation values on average compared with supramodal areas (linear mixed-effects regression: $\beta = -0.042$, 95% CI $[-0.044, -0.040]$, $t(499) = -49.9$, p<2 · 10$^{-16}$). These distributions demonstrate that sensorimotor isocortical regions exhibit more similarity overall on the basis of homologous gene expression than do supramodal regions.

While we found that sensorimotor isocortical areas in the mouse brain were more similar to human brain regions than supramodal areas, the distributions of maximal correlation do not speak to the neuroanatomical patterns of organization for these matches. To understand how the similarity patterns of mouse and human cortical subdivisions were organized, we used hierarchical clustering to cluster mouse and human isocortical regions on the basis of their similarity profiles in the average gene expression latent space (*Figure 5C*). This allows us to examine the similarity of regions to one another within and across brains at multiple levels simultaneously.

At a high level, we find a striking segregation of the mouse isocortex into one main cluster that corresponds to regions that are primarily engaged in sensorimotor processing and separate clusters of regions that are supramodal. All of the sensorimotor areas cluster together, but two supramodal areas also form part of this cluster: the posterior parietal association areas and the anterior cingulate cortex. The mouse sensorimotor cluster is characterized by high correlation values to human sensorimotor regions like the precentral gyrus, the cuneus, and the postcentral gyrus, as well as low correlation values to the piriform cortex and paraterminal gyrus. At this level of clustering, the remaining mouse supramodal subdivisions form three clusters. The retrosplenial area belongs to its own cluster, while the infralimbic and perirhinal areas cluster together. The similarity profile of the retrosplenial area is more similar to the sensorimotor cluster, and these two clusters are combined in the three-cluster solution. The remaining two mouse clusters are characterized by low correlations to the human cluster containing sensorimotor areas. This is especially true for the cluster containing the infralimbic and perirhinal areas.

On the human side, the four-cluster solution also features a sensorimotor cluster, which contains regions like the pre- and post-central gyri, the cuneus, and Heschl's gyrus. This cluster exhibits a high degree of similarity to the mouse sensorimotor cluster and low similarity to the mouse supramodal clusters. The isocortical regions not belonging to this cluster are split into three clusters. The majority of these remaining regions form a large cluster that contains areas like the cingulate gyrus and the frontal pole. The parolfactory gyri, parahippocampal gyrus, and temporal pole form a separate cluster that exhibits high correlation to the mouse ectorhinal, orbital, and prelimbic areas. Finally, the para-terminal gyrus and piriform cortex are clustered together and exhibit high similarity to the mouse infralimbic area and low similarity to the mouse sensorimotor cluster.

We additionally ran hierarchical clustering on the isocortical similarity matrix in the original homologous gene space. While the cluster annotations were not substantially different in this space, we observed that the Euclidean distances within and between clusters were smaller compared with the latent space clustering, further confirming that the perceptron classification approach improves the segregation of brain regions in the gene expression common space (*Figure 5D*).

Overall, we observe a greater degree of similarity between mouse and human cortical regions involved in basic sensorimotor processing compared with supramodal or association areas. This is in line with the large body of existing research that suggests that sensory and motor areas of the cortex are conserved across the brains of mammals. While sensorimotor areas exhibit a greater degree of similarity than supramodal areas, the neuroanatomical pattern of correspondences obtained using mouse-human homologous genes is not at the level of individual cortical areas. Still, using a clustering approach we identified clear distinctions in the patterns of similarity between sensorimotor and supra-modal areas, especially for regions in the mouse isocortex.

## Transcriptomic comparison of the mouse and human striatum

We have focused here on comparing mouse and human brain organization using transcriptomic data, with a latent space based on homologous genes as the common space between the two species. To date, common space comparisons between the mouse and human brain have only been performed using functional connectivity (*Balsters et al., 2020*; *Schaeffer et al., 2020*). As a case in point, *Balsters et al., 2020* compared mouse and human striatal organization using this measure. They found that the nucleus accumbens was highly conserved between mice and humans, and that voxels in the posterior part of the human putamen were significantly similar to the lateral portion of their mouse caudoputamen parcellation. Additionally, they report that 85% of voxels in the human striatum were not significantly similar to any of their mouse striatal seeds, and that 25% of human striatal voxels were significantly dissimilar compared with the mouse. These differences were understandable, as they involved parts of the human striatum that connected to parts of prefrontal cortex that have no known homologue in the mouse (*Neubert et al., 2014*). However, it is not necessarily the case that between-species differences in connectivity are associated with distinct architectonic or molecular signatures. Therefore, we investigated the patterns of similarity between the mouse and human striata on the basis of gene expression using the neural network latent space representations.

We first identified the striatal regions present in the Allen human dataset: the caudate, the putamen, and the nucleus accumbens. We evaluated the correlation between the microarray samples in these regions and every region in the mouse atlas. Based on these correlation values, we focused our analysis on the four mouse regions that were consistently the most similar across all latent spaces: the caudoputamen, the nucleus accumbens, the fundus of striatum, and the olfactory tubercle. For each of the human striatal regions, we then calculated the average correlation over the samples to each of the mouse targets. We examined the distribution of these average correlation values over the latent spaces (*Figure 6A*). We find that the human caudate and putamen consistently exhibit the strongest degree of similarity to the mouse caudoputamen. The median of distributions for the caudate-caudoputamen pairs and putamen-caudoputamen pairs is 0.93, with modal values of 0.92 and 0.94, respectively. All latent spaces return correlations greater than 0.85 for caudate-caudoputamen and putamen-caudoputamen pairs. Beyond this expected top match, the caudate and putamen both exhibit high similarity to the nucleus accumbens and the fundus of striatum, with mean correlation values of about 0.80. Neither of these target regions is consistently more similar to the mouse caudoputamen over all latent spaces.

While the similarity of the caudate and the putamen to the caudoputamen is unsurprising, the story is not as clear for the human nucleus accumbens. We find that the variance in correlation calculated over all mouse targets is much lower ($\sigma = 0.04$) compared with the equivalent variances for the caudate ($\sigma = 0.08$) and putamen ($\sigma = 0.08$), indicating less specificity to any one mouse striatal target. In particular, the human nucleus accumbens isn't as specifically similar to the mouse nucleus accumbens in the way that the caudate and putamen are similar to the caudoputamen. The mouse target distributions are right-shifted compared with those for the caudate and putamen, with median values of 0.89, 0.86, and 0.87 for the mouse nucleus accumbens, caudoputamen, and fundus of striatum, respectively. The human accumbens also exhibits a high degree of similarity to the mouse olfactory tubercle, the distribution of which is also right-shifted compared with the caudate and putamen.

Given the high correlation of the human caudate and putamen to the mouse caudoputamen, as well as the finding reported by Balsters et al. about the similarity of the lateral caudoputamen to the putamen, we were curious as to whether we could identify sub-regional patterns of similarity in the caudoputamen and other striatal regions using these gene expression data. To probe this question, we first examined the average latent space correlation between each voxel in the mouse striatum and every region in the human atlas. We created brain maps for the human regions that exhibited the highest mean correlation values, averaged over mouse striatal voxels: the caudate, the putamen, the nucleus accumbens, and the septal nuclei (*Figure 6B*). We find that voxels in the caudoputamen exhibit a homogeneous pattern of similarity to both the caudate and the putamen. On average, voxels in the caudoputamen have a correlation of 0.92 to the caudate and 0.91 to the putamen, with standard deviations of 0.05 and 0.06, respectively. The caudate and putamen are associated with correlations of at least 0.90 in 79 and 73% of caudoputamen voxels. A number of voxels are also highly similar to the human nucleus accumbens, with an average correlation value of 0.86 and 30% of voxels returning a correlation of at least 0.90. The caudoputamen voxels most similar to the nucleus accumbens lie

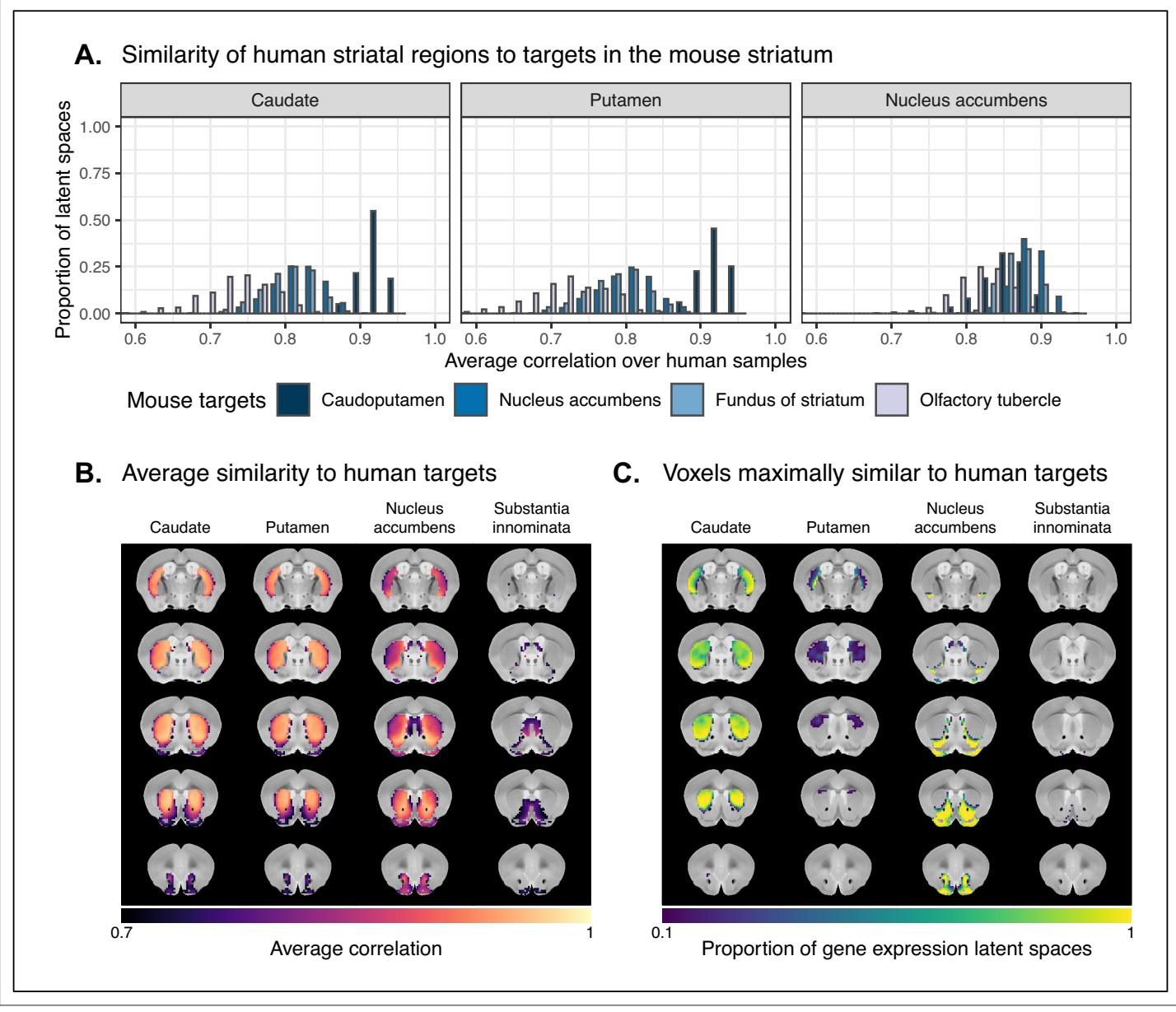

**Figure 6.** Similarity among mouse and human striatal regions. (**A**) Distributions over gene expression latent spaces of region-wise average correlation values for mouse and human striatal pairs. Human regions were chosen based on the Allen Human Brain Atlas ontology. Mouse target regions were chosen to be those with the highest average correlation values. (**B**) Latent space averaged correlations between voxels in the mouse striatum and human target regions. Target regions were selected based on the highest mean correlation across all striatal voxels. (**C**) Proportions of latent spaces in which mouse striatal voxels are maximally similar to human target regions.

The online version of this article includes the following source data for figure 6:

**Source data 1.** Correlations between human striatal samples and mouse striatal targets in all latent spaces, related to *Figure 6A*.

**Source data 2.** Average latent space correlations of mouse striatal voxels with human regions, related to *Figure 6B*.

**Source data 3.** Maximal correlations of mouse striatal voxels in all latent spaces (1 of 3), related to *Figure 6C*.

**Source data 4.** Maximal correlations of mouse striatal voxels in all latent spaces (2 of 3), related to *Figure 6C*.

**Source data 5.** Maximal correlations of mouse striatal voxels in all latent spaces (3 of 3), related to *Figure 6C*.

in the ventral-rostral part of the region. Of course, voxels in the mouse nucleus accumbens are also highly similar to the human nucleus accumbens, with an average of 0.89 and standard deviation of 0.06. While the human nucleus accumbens is the most strongly correlated region, a number of voxels also exhibit reasonably strong correlations to the substantia innominata and the amygdala. Indeed, 88% of voxels in the accumbens are correlated at a value of 0.7 or higher to the amygdala, and 57% of voxels pass this threshold for the substantia innominata.

We additionally examined the proportion of latent spaces in which each voxel in the mouse striatum was maximally similar to the human target regions (*Figure 6C*). As expected, we find that voxels in the caudoputamen are most often maximally similar to the human caudate and putamen, with 77% of voxels in the caudoputamen being maximally similar to the caudate or putamen in at least 95% of latent spaces, and 59% of voxels being maximally similar to one of those targets in all latent spaces. Interestingly, we observe the emergence of a continuous bilateral pattern of specifity to the caudate and putamen, with voxels in the rostral and lateral-caudal parts of the caudoputamen being maximally similar to the caudate in a high proportion of latent spaces. In contrast, while voxels in the medial-rostral part of the caudoputamen are often maximally similar to the caudate, they are also maximally similar to the putamen in some of latent spaces. This map highlights subtle differences in the similarity between caudoputamen voxels and the caudate or putamen. While this pattern distinguishes the two regions on the basis of which is the top match, individual voxels have very similar correlation values to the targets (*Figure 6B*), with a mean difference in correlation of only 0.01. Beyond the caudoputamen, we find that the accumbens and olfactory tubercle in the mouse are consistently similar to the human nucleus accumbens, with 84% of mouse accumbens voxels and 75% of olfactory tubercle voxels having the human accumbens as their top match in at least 80% of latent spaces. For those voxels below this threshold, the human regions that are most often the top match are the amygdala and the piriform cortex.

Overall, we observe a strong association between the mouse caudoputamen and both the human caudate and putamen. While we find a subtle pattern of specificity to either region among voxels in the caudoputamen on the basis of maximal similarity, the high degree of similarity in the correlation values to each region suggests that the majority of voxels in the caudoputamen are equally similar to the caudate and the putamen on the basis of the expression of mouse-human homologous genes. We also find that the nucleus accumbens is well conserved across species. However, the region also exhibits patterns of similarity that go beyond the simple one-to-one match. The human accumbens features similar correlation values to the mouse caudoputamen and fundus of striatum, in addition to the accumbens proper, with no sharp distinction between these regions. It also exhibits a larger degree of similarity to the mouse olfactory tubercle. This is also seen in the mouse striatum, where voxels in the accumbens and the olfactory tubercle map onto the human accumbens.

## Discussion

We have demonstrated how spatial transcriptomic patterns of homologous genes can be used to make quantitative comparisons between the mouse and human brain. We showed that using homologous genes as a common space allows one to easily identify coarse similarities in brain structures across species, but that more fine-scaled parcellations, such as at the level of cortical areas, are more complex. Despite this limitation, the approach still allows for a formal assessment of different patterns of between-species similarity in primary compared to supramodal regions, identifications of distinct clusters of cortical territories across species, and comparison of between-species similarities at the transcriptomic level to those observed using other modalities. We will discuss our observations in the context of the importance of the mouse as a model for human neuroscience below.

The abundance of neuroscience research performed using mice has resulted in a wealth of knowledge about the mouse brain. In the preclinical setting, mouse models are utilized with the intention of better understanding human neuropathology. For instance, in the context of autism spectrum disorders, a plethora of studies using mouse models have reported on the neurobiological and neuroanatomical phenotypes that arise from mutations at specific genetic loci (*Gompers et al., 2017*; *Horev et al., 2011*; *Pagani et al., 2021*). It is common for researchers involved in translational neuroscience to rely on findings of this kind to make inferences about the human disorder. The typical approach, which is to identify rough post-hoc correspondences between neuroanatomical ontologies, is not particularly comprehensive and is subject to confirmation bias. While it may be a reasonable starting

point for comparison, the true correspondence between the mouse and human brain is likely more complicated given the evolutionary distance between the two species. Although overall patterns of brain organization, including the general pattern of neocortical organization, are similar across all mammals, substantial differences are evident (*Ventura-Antunes et al., 2013*). To make matters worse, researchers from the different neuroscientific traditions often use distinct terminology, further complicating detailed information exchange. To address these problems, we sought to establish a first quantitative whole-brain comparison between the two species.

The expression of homologous genes provides an elegant way to define a common space for quantitative cross-species comparisons since it relies on homology at a deep molecular biological level. The approach is not without limitations, however. First, the acquisition of whole-brain transcriptomic data is labour-intensive, time-consuming, and invasive. These data sets cannot be generated easily, especially in the human, in which the process depends on the availability of post-mortem samples. As a result, the effective sample sizes are extremely limited in this domain. For instance, in the Allen mouse coronal in-situ hybridization data set used here, the brain-wide expression of each gene is sampled only once (barring a few exceptions). This constrains the types of analyses that are possible (e.g. null hypothesis significance testing) and largely limits the availability of replication data sets. That being said, new technologies, such as spatial transcriptomics, are gradually making it easier to acquire brain-wide gene expression data in less time and at lower cost (*Ortiz et al., 2020*; *Ståhl et al., 2016*; *Vickovic et al., 2019*). Second, the approach of relying on all available genes is subject to noise. To address this issue, *Myers, 2017* used a method of gene set selection to attempt to improve the correspondence between established mouse-human homologies. While this leads to improvement, it was only at the level of coarsely defined regions (e.g. cortex-cortex). Our approach, therefore, was to use supervised machine learning to create a latent common space based on combinations of homologous genes that can delineate areas within a single species.

This latent common space approach led to a substantial improvement in specificity of between-species comparisons. Nevertheless, it is evident that the first major distinction in gene expression patterns within a species, and the easiest identification of similarity across species, are at the coarse anatomical level of the major subdivisions of the vertebrate brain, such as the isocortex, cerebellar hemispheres and nuclei, and brain stem. All of these territories were present in the ancestral vertebrate brain (*Striedter and Northcutt, 2020*), and the ability to detect conserved transcriptomic signatures at this level is not surprising. Within such structures, such as the isocortex, our ability to make simple one-to-one correspondences decreased. This is partly because areas within a coarse structure have more similar transcriptomic profiles, but also likely due to the fact that a single area in one brain does not have a single correspondent in another, larger brain. In other words, regions in the brains of related species may exhibit one-to-many or many-to-many mappings. In our study, we found greater cross-species similarity between isocortical areas associated with sensorimotor processing than areas in supramodal isocortex. Primary areas, including the sensorimotor areas, are present in all mammals studied to date and likely part of the common ancestors of all mammals (*Kaas, 2011a*; *Krubitzer, 2007*). Although this common ancestor likely also had non-primary areas, it cannot be denied that association cortex expanded dramatically in primates and especially so in the human brain (*Chaplin et al., 2013*; *Mars et al., 2016b*). Again, the pattern found here of greater similarity in more conserved areas might reflect this evolutionary history. In that context it is interesting to note that some non-primary areas thought to be present in the common mammalian ancestor, such as cingulate and orbitofrontal cortex (*Kaas, 2011a*) showed relatively high correlation to human areas.

An advantage of the approach presented here is that it can, in principle, be applied to any aspect of brain organization. Beyond simply establishing whether areas are similar across species in a particular common space, comparing the results across common spaces established using different types of neuronal data can inform on which larger principles of organization are similar across brains (*Eichert et al., 2020*). This is illustrated here by the results of our striatal analysis. We found high similarity between the human caudate and putamen and mouse caudoputamen, with little differentiation within these regions in a single species. In contrast, *Balsters et al., 2020* demonstrated that human caudoputamen contains a distinct pattern of connectivity. At first sight, one could argue the results are in contrast. However, evolutionarily speaking, it is quite probable that an overall similar transcriptomic signature of the striatum can be accompanied by a distinct connectivity pattern to areas of the cortex present in only one of the two species. Indeed, this speaks to the different types of similarity that can

be studied, depending on which aspect of brain organization one is interested in. Although the human brain is much larger than the mouse brain and contains a number of cortical territories that have no homologue in the mouse brain (*Kaas, 2011b*; *Rudebeck and Izquierdo, 2022*), the similarity in transcriptomic signature mean that translations between the species is valid in many contexts. The supervised learning approach also provides interesting avenues for future research. For instance, rather than classifying all regions in the brain at once, separate models could be trained to classify regions belonging to different sub-trees in the neuroanatomical hierarchy (see *Figure 5—figure supplement 1* and *Figure 5—figure supplement 2*). This type of approach requires more exploration, however, such as where to split the hierarchy, how to optimize the classifiers for each sub-tree, and how to stitch all this information back together at the end in order to make comparisons between different sub-trees.

The power of a formal understanding of similarities and differences between brains at different levels of organization is evident. In fundamental neuroscience, it will help translate results from data types that cannot be obtained in humans to the human brain (*Barron et al., 2021*). In translational neuroscience, it will, in a negative sense, help establish the limits of the translational paradigm by showing which aspects of the human brain cannot be understood using the model species (*Liu et al., 2021*). In a positive sense, it will also help by establishing and improving our understanding of the many aspects in which the model and human brain do concur (*Mandino et al., 2021*). More ambitious still, it can provide a way in which highly diverse manifestations of certain disease syndromes (e.g. autism spectrum disorder) (*Grzadzinski et al., 2013*; *Simonoff et al., 2008*) and the availability of many distinct model strains (*Ellegood et al., 2015*), each hypothesized to capture a distinct aspect of a multi-dimensional clinical syndrome, can be related to one another. Ultimately, we believe that using the mapping of homologous gene expression between species can be an important part of building a transform that maps information obtained using mice to humans and vice versa.

## Materials and methods
### Mouse gene expression data
We used the adult mouse whole-brain in-situ hybridization data sets from the AMBA (*Lein et al., 2007*). Specifically, we used 3D expression grid data, that is, expression data aligned to the Allen Mouse Brain Common Coordinate Framework (CCFv3) (*Wang et al., 2020*) and summarized under a grid at a resolution of $200\mu m$. We downloaded the gene expression 'energy' volumes from both the coronal and sagittal in-situ hybridization experiments as a sequence of 32-bit float values using the Allen Institute's API (http://help.brain-map.org/display/api/Downloading+3-D+Expression+Grid+Data). These volumes were subsequently reshaped into 3D images in the Medical Image NetCDF (MINC) format. Origin, extents, and spacing were defined such that the image was RAS-oriented, with the origin at the point where the anterior commissure crosses the midline. The MINC images from the coronal and sagittal data sets were then processed separately using the Python programming language. The sagittal data set was first filtered to keep only those genes that were also present in the coronal set. Images were imported using the `pyminc` package, masked and reshaped to form an experiment-by-voxel expression matrix. We pre-processed this data by first applying a `log2` transformation for consistency with the human data set. For those genes associated with more than one in-situ hybridization experiment, we averaged the expression of each voxel across the experiments. We subsequently filtered out genes for which more than 20% of voxels contained missing values. Finally, we applied a K-nearest neighbours algorithm to impute the remaining missing values. The result of this pre-processing pipeline was a gene-by-voxel expression matrix with 3958 genes and 61,315 voxels for the coronal data set and a matrix with 3619 genes and 26,317 voxels for the sagittal data set.

### Human gene expression data
Human gene expression data was obtained from the AHBA (*Hawrylycz et al., 2012*). The data were downloaded from the Allen Institute's API (http://api.brain-map.org) and pre-processed using the `abagen` package in Python (https://abagen.readthedocs.io/en/stable/) (*Arnatkeviciute et al., 2019*; *Hawrylycz et al., 2012*; *Markello et al., 2021*). We used the microarray data from the brains of all six donors, each of which contains `log2` expression values for 58,692 gene probes across numerous

tissue samples. The pre-processing pipeline included probe selection using differential stability on data from all donors and intensity-based filtering of probes at a threshold of 0.5. The samples and genes were additionally normalized for each donor individually using a scaled robust sigmoid function. In practice, this pipeline was implemented using the `get_samples_in_mask` function from the `abagen` package. The remaining parameters were set to their default values. The output of the pre-processing pipeline was a gene-by-sample expression matrix with 15,627 genes and 3702 samples across all donors.

## Mouse atlases

We used a version of the DSURQE atlas from the Mouse Imaging Centre (*Dorr et al., 2008*; *Qiu et al., 2018*; *Richards et al., 2011*; *Steadman et al., 2014*; *Ullmann et al., 2013*), modified using the AMBA hierarchical ontology, which was downloaded from the Allen Institute's API. The labels of the DSURQE atlas correspond to the leaf node regions in the AMBA ontology, which allowed us to use the hierarchical neuroanatomical tree to aggregate and prune the atlas labels to the desired level of granularity. For the purposes of our analyses, we removed white matter and ventricular regions entirely. The remaining gray matter regions were aggregated up the hierarchy so that the majority of resulting labels contained enough voxels to be classified appropriately by the multi-layer perceptron. In doing so, we maintained approximately the same level of tree depth within a broad region (e.g. cerebellar regions were chosen at the same level of granularity). This resulted in a mouse atlas with 67 gray matter regions. We additionally generated an atlas with 11 broader regions for visualization and annotation purposes.

## Human atlases

We used the hierarchical ontology from the AHBA, which we obtained using the Allen Institute's API. We aggregated and pruned the neuroanatomical hierarchy to correspond roughly to the level of granularity obtained in our mouse atlas, resulting in 88 human brain regions. We additionally generated a set of 16 broad regions for visualization and annotation. White matter and ventricular regions were omitted entirely.

## Expression matrices and similarity matrices

We created the mouse and human gene-by-region expression matrices from the mouse gene-by-voxel and human gene-by-sample expression matrices. First, we intersected the gene sets in these matrices with a list of 3331 homologous genes obtained from the NCBI HomoloGene database (*NCBI Resource Coordinators, 2018*), resulting in 2835 homologous genes present in both the mouse and human expression matrices. We then annotated each of the human samples with one of the 88 human atlas regions, and each of the mouse voxels with one of the 67 mouse atlas regions, discarding white matter and ventricular entries in the process. These labeled expression matrices were subsequently normalized as follows: For each matrix, we first normalized each voxel/sample across all homologous genes using a z-scoring procedure to create a normalized gene expression signature for each voxel/sample. We then centered the distribution of expression signatures in gene space by subtracting the mean expression of each homologous gene over all voxels/samples. Finally, we generated the gene-by-region expression matrices by averaging the expression of every gene over the voxels/samples corresponding to each atlas region. Using these expression matrices, we generated the mouse-human similarity matrix by computing the Pearson correlation coefficient between all pairs of mouse and human regions.

## Gene enrichment analysis

We ran a gene enrichment analysis on the set of homologous genes obtained from the NCBI HomoloGene database. We first downloaded Gene Ontology data for biological process related modules from the Bader Lab at the University of Toronto (http://baderlab.org/GeneSets). These data include a gene set of 16,563 genes and a module set of 15757 biological process modules. Every module is associated with a subset of genes from the full gene set. For each module, we used a hypergeometric test to evaluate whether the homologous gene set was over-represented in the module subset, compared with the full gene set. The resulting p-values were adjusted for multiple comparisons using the false-discovery rate method (*Benjamini and Hochberg, 1995*). A total of 938 modules were found

to be significant at a threshold of 0.001. The surviving modules were ordered according to their p-values and written out to a comma-separated values data file (*Supplementary file 1*). This analysis was carried out using the `tmod` package in the R programming language.

## Multi-layer perceptron classification and latent space

To improve the resolution of mouse-human neuroanatomical matches, we performed a supervised learning approach, wherein we trained a multi-layer perceptron neural network to classify 67 mouse atlas regions from the expression values of 2835 homologous genes. We chose a model architecture in which each layer of the network was fully connected to previous and subsequent layers. To optimize the hyperparameters, we implemented an ad hoc cross-validation procedure that took into account the fact that the majority of genes in the coronal AMBA data set are sampled only once over the entire mouse brain. The procedure involved a combination of the coronal data set and the sagittal in-situ hybridization data sets. For the sagittal data set, we used the expression matrix described above. However, we used a modified version of the coronal expression matrix. This matrix was generated using the pipeline described above with the following modifications: (1) We applied the *unilateral* brain mask from the sagittal data set to the coronal images in order to have the same spatial extent, and (2) we did not aggregate the expression of multiple in-situ hybridization experiments for those genes in the coronal set that were measured more than once. We then filtered these experiment-by-voxel expression matrices according to the list of mouse-human homologous genes, as well as the human sample expression matrix. We also annotated the voxels in each of the expression matrices with one of the 67 regions in the mouse atlas. Our validation procedure then involved iterative construction of training and validation sets by sampling gene experiments from either the coronal or sagittal matrices. For every gene in the homologous set, we first determined whether that gene was associated with more than one experiment in the coronal matrix. If this was the case, we randomly sampled one of those experiments for the training set and one of the remaining experiments for the validation set. If the gene was associated with only one experiment in the coronal set, we randomly sampled either the coronal or sagittal experiment for the training set and the other for the validation set. Once the training and validation sets were generated, they were normalized using the procedure described above. We then optimized the neural network using the training set and evaluated its performance on the validation set. We repeated this construction, training, and validation procedure five times for every combination of hyperparameters.

Using this validation approach, we tuned the number of hidden layers in the network, the number of hidden units per hidden layer, the amount of weight decay, the maximum learning rate, and the optimization method. The values we sampled were as follows:

Number of hidden layers: 3, 4, 5
Number of hidden units: 200, 500, 1000
Weight decay: $0, 10^{-6}, 10^{-3}$
Maximum learning rate: $10^{-5}, 10^{-4}, 10^{-3}, 10^{-2}, 10^{-6}, 10^{-1}$
Optimizer: SGD, `AdamW`

All models were trained over 200 epochs using a one-cycle learning rate policy. The activation function used in the forward pass was the rectified linear unit, and the loss function was the negative log-likelihood loss. We found that the best-performing model had 3 hidden layers, 200 neurons per layer, and no weight decay. It was optimized using the AdamW optimization algorithm (*Loshchilov and Hutter, 2019*) with a maximum learning rate of $10^{-5}$. This model returned an average loss of 0.215 on the training sets and of 1.224 on the validation sets. The average training classification accuracy was 0.936, and the validation accuracy was 0.597.

Using the optimal hyperparameters, we trained the multi-layer perceptron on the full bilateral coronal voxel-wise expression matrix. We used the trained network to generate the latent gene expression space. To extract the appropriate transformation, we removed the predictive output layer and soft-max transformation from the network architecture. The resulting architecture returns the 200 hidden units in the third hidden layer as the output of the model. To create the latent space data representations, we applied this network to the mouse and human regional and voxel-/sample-wise expression matrices. The resulting matrices have 200 columns corresponding to the hidden units and rows corresponding to the number of regions, voxels, or samples in the mouse and human matrices. This process was repeated 500 times to generate 500 latent spaces.

These models were implemented in Python using PyTorch (https://pytorch.org) and the skorch package (https://skorch.readthedocs.io/en/stable/).

## Multi-layer perceptron feature importance

We used integrated gradients to evaluate the contribution of different genes in the classification of mouse atlas labels. Since the homologous gene inputs contribute to the classification of distinct labels in different ways, we examined the feature attributions for three regions: the caudoputamen, the primary motor area, and the infralimbic area. Using the trained multi-layer perceptron, we computed integrated gradients for each of these three regions. We then averaged the values over all input voxels for each gene, resulting in a vector of gene attributions for each of the three example regions. This process was repeated for 200 training runs of the neural network. We then averaged the gene importance vectors of each region over all training runs to get a summary of gene importance. This process was implemented using the IntegratedGradients function from the captum package in Python (https://captum.ai/).

## Statistical modeling

To quantify the improvement in the mouse-human matches when using the latent spaces versus the original gene expression space (*Figures 3 and 4*), we used a set of logistic regression models to estimate the probability that the rank difference was less than or equal to zero. To estimate the overall improvement due to the latent spaces, we created a binary variable to encode whether the average rank difference over latent spaces for each region met the success criterion. This variable was then used as our target in a logistic regression with no regressors. Once the model was fit, we applied the logistic function to the intercept parameter estimate to get the corresponding estimate for the Bernoulli probability, $p_B$. This transformation was also applied to the bounds on the variance estimate for the intercept to get the corresponding confidence interval. Using the estimated Bernouilli probability, we calculated the corresponding number of successes, $k$. We then evaluated the probability of obtaining at least $k$ successful outcomes under the null binomial distribution, $B(n, 0.5)$. The parameter $n$ was taken to be the number of brain regions under consideration. We additionally applied this approach on a region-wise basis to evaluate the likelihood of a region seeing improvement in the latent spaces. In this case, the null distribution was $B(500, 0.5)$ for each region. The resulting p-values were adjusted for multiple comparisons using the false-discovery rate method (*Benjamini and Hochberg, 1995*). These models were implemented using the glm function from the stats package in the R programming language.

In our comparison of sensorimotor and supramodal cortical regions (*Figure 5*), we used linear models to evaluate the impact of cortex type on maximal correlation values. In the first instance, we computed each region's average maximal correlation over all latent spaces. We then regressed those average values against a binary variable indicating whether the regions were sensorimotor or supramodal. Here we used a simple linear regression. In the second instance, for each latent space we computed average maximal correlation values for sensorimotor regions and supramodal regions. We then regressed these average values against a binary variable as described above. In this case, lm function from the stats package, while the linear mixed-effects regression was implemented using the lmer function from the lme4 package. The lmerTest package was used to estimate the degrees of freedom in the mixed-effects model and perform hypothesis testing.

## Acknowledgements

We thank C Hammill, DJ Fernandes, E Anagnostou, BJ Nieman, and E Sibille for providing advice and for interesting conceptual discussions. This study was supported by the Canadian Institutes of Health Research (doctoral funding and foreign study award for AB), the National Institutes of Health (grant 5R01HD100298), and the E P A Cephalosporin Fund. The Wellcome Centre for Integrative Neuroimaging is supported by core funding from the Wellcome Trust (203139/Z/16/Z).

## Additional information

### Funding

| Funder | Grant reference number | Author |
| --- | --- | --- |
| Canadian Institutes of Health Research | GSD-165737 | Antoine Beauchamp |
| Wellcome Trust | 203139/Z/16/Z | Rogier B Mars Jason P Lerch |
| University of Oxford | E P A Cephalosporin Fund | Rogier B Mars |
| National Institutes of Health | 5R01HD100298 | Armin Raznahan Jason P Lerch |
| Canadian Institutes of Health Research | FSS-167844 | Antoine Beauchamp |

The funders had no role in study design, data collection and interpretation, or the decision to submit the work for publication. For the purpose of Open Access, the authors have applied a CC BY public copyright license to any Author Accepted Manuscript version arising from this submission.

### Author contributions

Antoine Beauchamp, Conceptualization, Data curation, Software, Formal analysis, Funding acquisition, Validation, Investigation, Visualization, Methodology, Writing – original draft, Project administration, Writing – review and editing; Yohan Yee, Conceptualization, Data curation, Software, Methodology, Writing – review and editing; Ben C Darwin, Software, Validation, Methodology, Writing – review and editing; Armin Raznahan, Data curation, Writing – review and editing; Rogier B Mars, Jason P Lerch, Conceptualization, Resources, Supervision, Funding acquisition, Investigation, Methodology, Writing – original draft, Project administration, Writing – review and editing

### Author ORCIDs

Antoine Beauchamp http://orcid.org/0000-0002-0008-7471
Yohan Yee http://orcid.org/0000-0001-7083-1932
Ben C Darwin http://orcid.org/0000-0001-8689-046X
Armin Raznahan http://orcid.org/0000-0002-5622-1190
Rogier B Mars http://orcid.org/0000-0001-6302-8631

### Decision letter and Author response

Decision letter https://doi.org/10.7554/eLife.79418.sa1
Author response https://doi.org/10.7554/eLife.79418.sa2

## Additional files

### Supplementary files
• Supplementary file 1. Biological modules enriched in the homologous gene set.
• MDAR checklist

### Data availability
The Allen Mouse Brain Atlas and Allen Human Brain Atlas data sets are openly accessible and can be downloaded from the Allen Institute's API (http://api.brain-map.org). This manuscript and all figures were generated programmatically using R Markdown (https://rmarkdown.rstudio.com) and (https://www.latex-project.org). All of the code and additional data needed to generate this analysis, including figures and manuscript, is accessible at GitHub, (copy archived at swh:1:rev:0ad9c547e18e8ca5d-08872cbecb9f729a4b8b62b; *Beauchamp, 2022*).

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
