## [Editor Report]

This important work develops new methods for aligning measures of brain-wide gene expression in the mouse and human brains. It presents compelling evidence in support of both conserved and species-specific transcriptional patterns. The work will be of interest to neuroscientists and geneticists interested in the molecular correlates of brain evolution.

---

## [Decision Letter]

**Decision letter after peer review:**

Thank you for submitting your article "Whole-brain comparison of rodent and human brains using spatial transcriptomics" for consideration by *eLife*. Your article has been reviewed by 2 peer reviewers, and the evaluation has been overseen by a Reviewing Editor and Kate Wassum as the Senior Editor. The following individual involved in the review of your submission has agreed to reveal their identity: Bratislav Misic (Reviewer #2).

Essential revisions:

Each of the Reviewers has raised some specific points that require further attention or analysis in their public review and recommendations for authors. Please provide a point-by-point response to each of these. We especially ask that you well address Reviewer 1 recommendations to authors points 2 and 3.

In your revision, If you have not already done so, please ensure your manuscript complies with the *eLife* policies for statistical reporting: https://reviewer.elifesciences.org/author-guide/full "Report exact p-values wherever possible alongside the summary statistics and 95% confidence intervals. These should be reported for all key questions and not only when the p-value is less than 0.05."

*Reviewer #1 (Recommendations for the authors):*

From the methodological point of view, this study is well-executed and the specific questions/suggestions are presented below.

1. Expression patterns across broad anatomical divisions such as the human cortex, subcortex, brainstem, and cerebellum demonstrate substantial differences. Similar tendencies are also observed in the mouse brain, where differences between neocortical and other brain areas tend to be much stronger compared to the differences within these divisions. The analyses presented in this work are performed on the combined datasets covering the whole brain and the resulting similarity metrics appear to be significantly skewed to the right with values broadly ranging from 0.7-1. Could the authors please comment if these transcriptional differences between broad anatomical divisions may attenuate/diminish the potential differences within these structures, e.g. within cortex/neocortex/subcortex/cerebellum? It might be interesting to expand the analyses by analyzing each anatomical division independently in order to disentangle more subtle transcriptional similarities/differences between species.

2. Currently, in the description of the processing of AHBA data there is no mention of within-donor normalization prior to data aggregation. It has been previously shown that samples acquired from the same donor tend to cluster together rather than reflecting anatomical divisions of the brain when samples across 6 brains are combined. Based on the current documentation, samples from all 6 brains are first aggregated into a sample x gene matrix and only then normalized for every gene across samples. This type of normalization retains expression differences between different donor brains and can bias the resulting sample x gene and region x gene datasets as well as subsequent analyses. Markello et al., (2021) have recently shown that within-donor data normalization is the most influential step in AHBA data processing, therefore, I suggest revisiting this data processing step. Also, could the authors comment on the choice of mean expression level subtraction for within-sample/region normalization rather than the standard z-score normalization?

3. Does the latent gene space method allows the identification of genes that are most informative in region identification? Could the authors provide some comments in the manuscript?

4. Some formal statistical evaluations should be presented when performing comparisons. For example, but not limited to, comparing maximal correlational values between sensimotor and supramodal areas (lines 277-280, Figure 5B).

References

Markello, R. D., Arnatkeviciute, A., Poline, J.-B., Fulcher, B. D., Fornito, A., and Misic, B. (2021). Standardizing workflows in imaging transcriptomics with the abagen toolbox. *eLife*, 10, e72129. https://doi.org/10.7554/*eLife*.72129

*Reviewer #2 (Recommendations for the authors):*

I think the manuscript is very polished as-is. I have a number of questions/suggestions that should be considered optional:

1) Line 61: "the connections of a brain region tend to be unique". I know exactly what the authors mean (each brain region has a unique/specific connectivity profile), but the sentence could perhaps be clearer.

2) Why use a multi-layer perceptron to map homologues, as opposed to a more interpretable, SVD-based method, such as PLS or CCA?

3) It is still not entirely clear to me how well the perceptron performs in the more conventional, global sense – is there a final, cross-validated accuracy? Is this accuracy significantly greater than what would be expected by chance?

4) In most of the analyses, there is a clear distinction between the cortex and cerebellum, which should then be expected to drive the configuration of the latent spaces. Have the authors attempted to perform the analysis using cortex only?

5) Do the authors have a sense of what biological pathways the homologous genes are involved in?

---

## [Author Response]

Essential revisions:Each of the Reviewers has raised some specific points that require further attention or analysis in their public review and recommendations for authors. Please provide a point-by-point response to each of these. We especially ask that you well address Reviewer 1 recommendations to authors points 2 and 3.In your revision, If you have not already done so, please ensure your manuscript complies with the eLife policies for statistical reporting: https://reviewer.elifesciences.org/author-guide/full "Report exact p-values wherever possible alongside the summary statistics and 95% confidence intervals. These should be reported for all key questions and not only when the p-value is less than 0.05."Reviewer #1 (Recommendations for the authors):From the methodological point of view, this study is well-executed and the specific questions/suggestions are presented below.1. Expression patterns across broad anatomical divisions such as the human cortex, subcortex, brainstem, and cerebellum demonstrate substantial differences. Similar tendencies are also observed in the mouse brain, where differences between neocortical and other brain areas tend to be much stronger compared to the differences within these divisions. The analyses presented in this work are performed on the combined datasets covering the whole brain and the resulting similarity metrics appear to be significantly skewed to the right with values broadly ranging from 0.7-1. Could the authors please comment if these transcriptional differences between broad anatomical divisions may attenuate/diminish the potential differences within these structures, e.g. within cortex/neocortex/subcortex/cerebellum? It might be interesting to expand the analyses by analyzing each anatomical division independently in order to disentangle more subtle transcriptional similarities/differences between species.

The idea of separately classifying subtrees of the hierarchical ontology (e.g.isocortex, cerebellum) is a good idea, and one that we’d considered previously. Given the difficulty of discriminating between cerebellar regions using transcriptomic data, we believe that this approach would be most beneficial in the isocortex. To that end, we trained the multi-layer perceptron to classify the 19 regions in the AMBA ontology that make up the mouse isocortex, and then generated 500 latent spaces in the way we’ve described previously. In this case, however, we only transformed mouse and human isocortical regions into the latent spaces, since the network was only trained to classify isocortical regions. Given this reduced set of regions, we focused on examining the impact of the isocortical latent spaces on the outcomes that were specific to isocortical regions.

We first generated a version of Figure 4 that included only those mouse isocortical regions with established neuroanatomical homologues. Here we find that in terms of averages, the original set of latent spaces and the isocortical-specific latent spaces perform equally well for most of the regions surveyed. The exception is the visual areas, which feature an improve mouse-human correspondence in the isocortical-specific spaces. Another salient feature is that the variance in the rank of the canonical pairs is smaller in the isocortical-specific latent spaces.

**Author response image 1. sa2fig1:** 

In addition to this analysis, we repeated the clustering analysis from Figure 5 using the isocortical latent spaces.We find striking differences here. The first is that supramodal regions exhibit higher maximal correlation values than sensorimotor regions on average (panels A and B). Another important change is that average latent space similarity matrix exhibits fewer pairs of regions with high correlations (panel C). This in turn changes the clustering. The mouse sensorimotor regions no longer form a single cluster but are split off into three clusters. A sensorimotor cluster mostly remains on the human side, though the precentral gyrus is split off into a different cluster. Interestingly, the scree plot in panel D suggests that the optimal clustering in this space might be as high as 10 clusters.

While we find the subtree classification approach interesting, we don’t believe that we see enough improvement in this preliminary analysis to justify switching approaches at this stage. The global classification task performs well and offers a simple way to evaluate comparisons between different regions across the entire brain. Tackling the subtree classification approach in full is an interesting avenue for future work but would require more exploration. In particular, the latent spaces obtained from the classification of different subtrees cannot trivially be stitched together. This renders it difficult to make comparisons between different broad regions of the brain, e.g. isocortical regions and sub-cortical regions. Additional considerations include where to split the anatomical hierarchy for classification, and how to optimize networks for each sub-division. We’ve added the following sentences in the discussion (lines 660-666):

“The supervised learning approach also provides interesting avenues for future research. For instance, rather than classifying all regions in the brain at once, separate models could be trained to classify regions belonging to different sub-trees in the neuroanatomical hierarchy. This type of approach requires more exploration however, such as where to split the hierarchy, how to optimize the classifiers for each sub-tree, and how to stitch all this information back together at the end in order to make comparisons between different sub-trees.”

2. Currently, in the description of the processing of AHBA data there is no mention of within-donor normalization prior to data aggregation. It has been previously shown that samples acquired from the same donor tend to cluster together rather than reflecting anatomical divisions of the brain when samples across 6 brains are combined. Based on the current documentation, samples from all 6 brains are first aggregated into a sample x gene matrix and only then normalized for every gene across samples. This type of normalization retains expression differences between different donor brains and can bias the resulting sample x gene and region x gene datasets as well as subsequent analyses. Markello et al., (2021) have recently shown that within-donor data normalization is the most influential step in AHBA data processing, therefore, I suggest revisiting this data processing step. Also, could the authors comment on the choice of mean expression level subtraction for within-sample/region normalization rather than the standard z-score normalization?

Our original pre-processing pipeline was built before the release of the *abagen* Python package, which implements the pipeline options described in Markello et al. 2021. That being said, we recognize the importance of using standardized tools, and so we’ve modified the human pre-processing pipeline such that it uses the *abagen* toolkit. Compared with our original pipeline, this new pipeline implements probe selection via differential stability, as well as within-donor normalization of both samples and genes using a scaled robust sigmoid function. We have updated the subsection “Human gene expression data” in the “Materials and methods” section to reflect these changes (lines 705-716):

“Human gene expression data was obtained from the Allen Human Brain Atlas (Hawrylycz et al., 2012). The data were downloaded from the Allen Institute's API (http://api.brain-map.org) and pre-processed using the `abagen` package in Python (https://abagen.readthedocs.io/en/stable/) (Arnatkeviciūtė et al., 2019; Hawrylycz et al., 2012; Markello et al., 2021). We used the microarray data from the brains of all six donors, each of which contains `log2` expression values for 58692 gene probes across numerous tissue samples. The pre-processing pipeline included probe selection using differential stability on data from all donors and intensity-based filtering of probes at a threshold of 0.5. The samples and genes were additionally normalized for each donor individually using a scaled robust sigmoid function. In practice, this pipeline was implemented using the `get_samples_in_mask` function from the `abagen` package. The remaining parameters were set to their default values. The output of the pre-processing pipeline was a gene-by-sample expression matrix with 15627 genes and 3702 samples across all donors.”

Using this updated pipeline to pre-process the human expression data, we re-generated all downstream aspects of the analysis, including 500 new latent spaces. Notably, using this updated pipeline, our subset of mouse-human homologous genes now contains 2835 genes rather than the original 2624. This value has been updated in the manuscript. We updated all figures to examine the impact of the new human pipeline on the outcomes.

In Figure 1, we observe a slight increase in contrast in the similarity matrices (panel A), particularly for mouse regions in the cortical subplate, olfactory areas, and hippocampal formation.

In Figure 2, we also see a slight increase in contrast in the similarity matrix (panel C).

In Figure 3, panel B, we see some minor tweaks to the region-wise distributions. The biggest change occurs in the cerebellar regions, where we see some improvement (i.e. distributions have moved left) when using the *abagen* pre-processing pipeline, compared with our original pipeline.

In Figure 4, panel A, we see some worsening of the ranks in the initial homologous gene expression space when using the updated pre-processing pipeline compared with our original pipeline (e.g. claustrum, piriform area, subiculum, primary motor area). However, the ranks are improved in the latent spaces as desired. Note that in the figure at submission, some of the error bars in panel A were erroneously being discarded by the plotting routine since they fell outside of the range of the x-axis (e.g. claustrum, anterior cingulate area). This has been resolved in the updated figure. In panel B., we see an improvement in the proportions in the hippocampal formation when using the new processing pipeline.

In Figure 5, we see some minor shuffling of the order of isocortical regions in panel A. We also see a slight increase in the separate between the medians of the distributions in panel B. In panel C, the mouse clusters are unchanged. However, we see the emergence of a sensorimotor cluster on the human side. Sensorimotor regions that we present in the large cluster in the original figure (e.g. precentral gyrus, Heschl’s gyrus) are now clustered with the somatosensory and visual regions. This cluster is characterized by high similarity to the mouse sensorimotor cluster. In panel D, we find that the cluster separations have increased when using the updated pre-processing pipeline.

In Figure 6, panel A, the distributions of caudate-caudoputamen and putamen-caudoputamen correlation values are shifted slightly to the left in the update figure. They are also slightly wider. This widening is also seen for the distributions of the human nucleus accumbens. In panel B, the voxel-wise correlations to the caudate and putamen are slightly lower in absolute value, but the spatial pattern of correlation remains unchanged. In panel C, the proportions are different for the caudate and putamen. In the updated figure, we find that voxels in the mouse caudoputamen are most consistently maximally similar to the human caudate. The distinction between the caudate and putamen that we saw in the original figure is mostly subdued. While hints of it remain in the dorsal and caudal parts of the caudoputamen, these voxels are still mostly maximally similar to the caudate, rather than the putamen.

Overall, we find that pre-processing the human data using the *abagen* package, including within-donor normalization, doesn’t hugely impact the outcomes of the study. The most important change is the emergence of a human sensorimotor cluster in the isocortical analysis.

Note that these changes in the figures are associated with changes to the quantitative statements reported in the discussion for each figure. These are outlined below in our response to recommendation #4 from reviewer #1.

Finally, to address our choice of normalization for the expression matrices: Upon revision of the manuscript and code, we found that we incorrectly described the normalization procedure in our original submission. The correct normalization procedure, along with our reasoning, is as follows: We first performed a z-score normalization for each voxel or sample across all homologous genes to generate a normalized gene expression signature for each voxel/sample. While this is likely good enough on its own, we wanted to ensure that the point-cloud distributions for both the mouse and human data sets were centered at the origin in the gene expression space, rather than existing in separate domains. To ensure this for each species, we subtracted the mean value for each gene across all voxels/samples.The result is a set of voxel-wise/sample-wise gene expression signatures, centered at the origin in gene expression space.

The manuscript has been adapted accordingly (lines 743-748):

“These labelled expression matrices were subsequently normalized as follows: For each matrix, we first normalized each voxel/sample across all homologous genes using a z-scoring procedure to create a normalized gene expression signature for each voxel/sample. We then centered the distribution of expression signatures in gene space by subtracting the mean expression of each homologous gene over all voxels/samples.”

3. Does the latent gene space method allows the identification of genes that are most informative in region identification? Could the authors provide some comments in the manuscript?

The relative importance of input genes for the classification of voxels into atlas regions, called feature importance or feature attributions, can be obtained using integrated gradients (Sundararajan et al., 2017). This method can be implemented using the *captum* (https://captum.ai/) package in Python. Using this toolkit, we can identify the relative importance of all genes in the classification of any given single label (e.g. caudoputamen), but not the classification of all labels at once. A comprehensive characterization of the importance of all input genes for all 67 mouse atlas labels across all 500 latent spaces is beyond the scope of this study. However, we have applied the method to generate feature attributions for the classification of three labels: the caudoputamen, the primary motor area, and the infralimbic area. Using *captum*, we computed integrated gradients for each of these three labels for 200 training runs of the perceptron. For each target label of interest, we averaged the importance of each gene over the 200 training iterations to get an average measure of importance over latent spaces. These results are summarized in a new supplemental figure, which is tied to figure 2.

We discuss this additional figure in the section titled “A latent gene expression space improves the resolution of mouse-human associations” (lines 219-232):

“Although the neural network and associated latent space do not directly provide information about which genes are most important for the classification of specific mouse atlas labels, this type of information can be derived from the model using attribution methods such as integrated gradients (Figure 2-—figure supplement 1)(Sundararajan et al., 2017). Each brain region in the classification task is associated with the input genes in different ways, such that there isn't a single weighting of gene importance for the entire model. While most genes contribute to the classification of any given label in some capacity, it is often the case that the network relies on a reduced subset of genes to arrive at a decision. For example, the genes *Prrg2* and *Cd4* were found to be most influential for the classification of the caudoputamen, when the feature attributions were averaged over all training runs. In contrast, *Rfx4* and *Glra3* were the most influential for the classification of the primary motor area. In some cases, the spatial expression pattern of the gene clearly shows a demarcation of the region of interest (e.g. *Cd4*), but this is not always the case, nor is it necessary, as the network learns from the entire gene expression signature of all voxels.”

We also included a new subsection in “Materials and methods”, titled “Multi-layer perceptron feature importance” (lines 821-830):

“We used integrated gradients to evaluate the contribution of different genes in the classification of mouse atlas labels. Since the homologous gene inputs contribute to the classification of distinct labels in different ways, we examined the feature attributions for three regions: the caudoputamen, the primary motor area, and the infralimbic area. Using the trained multi-layer perceptron, we computed integrated gradients for each of these three regions. We then averaged the values over all input voxels for each gene, resulting in a vector of gene attributions for each of the three example regions. This process was repeated for 200 training runs of the neural network. We then averaged the gene importance vectors of each region over all training runs to get a summary of gene importance. This process was implemented using the `IntegratedGradients` function from the `captum` package in Python (https://captum.ai/).”

4. Some formal statistical evaluations should be presented when performing comparisons. For example, but not limited to, comparing maximal correlational values between sensimotor and supramodal areas (lines 277-280, Figure 5B).

We initially avoided relying on formal statistical evaluations due to the absence of biologically relevant sources of variance in the data sets, i.e. variation in mouse-human correlation values arising from variation in transcriptomic maps. Since all donors were aggregated on the human side, and each gene was sampled only once in the coronal data set on the mouse side, the sample size is effectively n = 1.

The resampling of the latent spaces does introduce variance and this can be used to make more formal statistical statements. The caveat here however is that we have complete control over the relevant sample size (the number of latent spaces), and so metrics like p-values are meaningless, since the sample size can be made arbitrarily high at low cost.

That being said, we formalized some of the quantitative statements made with respect to the analyses using statistical models. Note that the values cited below reflect the changes resulting from the updated human pre-processing pipeline, as discussed in our response to recommendation #2 from reviewer #1.

For Figure 3, panels B and C, we used binomial likelihood models to quantify 1. The probability that any given region would see improvement on average in the latent spaces, and 2. The probability that any given latent space would return an improvement, for each individual region. While the resulting estimate of the Bernoulli probability in these models is equivalent to the proportions cited in the original manuscript, the models additionally return a confidence interval around this value. The relevant text in the section “A latent gene expression space improves the resolution of mouse-human associations” has been updated to reflect these changes (lines 266-316):

“Examining the structure-wise distributions of these rank differences, we found that for the majority of regions in our mouse atlas, the classification approach resulted in either an improvement in the amount of locality within a broad region, or no difference from the original gene space (Figure 3, B and C). We quantified the improvement overall by fitting a logistic regression model with no predictors to the mean rank differences of each of the atlas regions. We considered the success condition for the Bernoulli trials to be a mean rank difference less than or equal to zero. The model estimate for the Bernoulli probability – which we denote pB to distinguish from the p-value p – was pB = 0.78 with a 95% confidence interval of [0.66,0.86]. In other words, 52 of the 67 brain regions saw an improvement on average when using the latent spaces. The probability of obtaining at least as many successes as this under the null model, i.e. a binomial distribution with pB = 0.50 and n = 67, is p = 8.64 · 10−7. We additionally evaluated the same kind of logistic regression on a region-wise basis to quantify how often the latent spaces resulted in an improvement for individual brain regions (Figure 3C). We found that for 46 regions (69%), the model estimated the probability to be at least at high as pB = 0.95. While confidence intervals varied around this estimate, the range between the upper and lower bound was only ever as high as 0.04. For 53 of the 67 regions (79%), the q-values, i.e. p-values adjusted for multiple comparisons, were effectively null, with the largest being q = 3.77 · 10−16. Of the remaining 14 regions, 13 had q-values equal to 1 and one region, the periacqueductal gray, had a q-value of q = 0.854. The regions with the smallest estimates for the Bernouilli probabilities are the dentate gyrus (pB = 0.0, no variance, q = 1), the striatum ventral region (pB = 0.016, 95% CI [0.008, 0.032], q = 1), and the lateral septal complex (p = 0.016, 95% CI [0.008,0.032], q = 1). The remaining regions with q = 1 are all subcortical and fall under the broad subdivisions of cerebral nuclei, olfactory areas, interbrain, midbrain, pons, medulla, and cerebellar nuclei. Beyond this binary measure of improvement, some regions exhibited a large range of differences in rank over the various latent spaces. In particular regions like the main olfactory bulb (mean rank difference of μ = 10, 95% CI [−12, 33]) and (accessory olfactory bulb μ = 9, 95% CI [−13,31]) exhibit a substantial degree of variance. Other than these two areas, regions within the olfactory areas (e.g. piriform area) were among those that benefited the most from the classification approach, showing improvement in all sampled latent spaces, with all Bernouilli probability estimates equal to 1 and all q-values equal to 0. While the effects, i.e. rank differences, are smaller, the similarity profiles of regions belonging to the isocortex and cerebellar cortex also saw an improvement in locality. All models for isocortical areas returned Bernouilli probability estimates greater than pB = 0.85 and q-values that were at most q = 1.35·10−67. Moreover, 9 of the 19 isocortical regions were improved in all latent spaces, i.e. pB = 1. Brain regions belonging to the cerebellar cortex saw similar improvement. In contrast, regions belonging to the cerebral nuclei, the diencephalon, midbrain and hindbrain did not see much improvement in this new common space, with an average Bernouilli probability estimate of pB = 0.36 for this subset. Other than the caudoputamen (pB = 0.99, 95% CI [0.97, 1.00], q = 1.35 · 10−139), the superior colliculus (pB = 0.90, 95% CI [0.87, 0.92], q = 9.82 · 10−81), and the inferior colliculus (pB = 0.75, 95% CI [0.71, 0.78], q = 3.12 · 10−30), all regions in this subset return q-values equal to 1. For many such regions the degree of locality appears to be worse in this space, though only by a small number of ranks, e.g. striatum ventral region (mean rank difference of μ = 4, 95% CI [1, 7]) and lateral septal complex (μ = 6, 95% CI [0, 11]). Indeed, computing the average rank difference over this subset of regions across all latent spaces, we find μ = 2 with 95% confidence interval [−5, 8]. These results demonstrate that the supervised learning approach used here can improve the resolution of neuroanatomical correspondences between the mouse and human brains, though the amount of improvement varies over the brain. Regions that were already well-characterized using the initial set of homologous genes (e.g. subcortical regions) did not benefit tremendously, but numerous regions in the cortical plate and subplate, as well as the cerebellum, saw an improvement in locality in this new common space.”

We have also updated the caption to Figure 3 (lines 318-332).

We applied the same kind of binomial likelihood models to the rank comparisons in Figure 4. The manuscript has been updated to reflect these changes (lines 333-368):

“While the supervised learning approach improved our ability to identify matches on a finer scale for a number of brain regions, this does not necessarily mean that those improved matches are biologically meaningful. The second criterion for evaluating the performance of the neural network addresses whether this improvement in locality captures what we would expect in terms of known mouse-human homologies. To this end, we examined the degree of similarity between established mouse-human neuroanatomical pairs, both in the initial gene expression space and in the set of latent spaces. We began by establishing a list of 36 canonical mouse-human homologous pairs on the basis of common neuroanatomical labels in our atlases. For each of these regions in the mouse brain, we compared the rank of the canonical human match in the rank-ordered similarity profiles between the latent spaces and the original gene expression space (Figure 4A). The lower the rank, the more similar the canonical pair, with a rank of 1 indicating maximal similarity. As described above, we evaluated the overall performance of the classification approach by running a logistic regression using the average latent space rank difference over all regions in our subset. Here we find an estimated Bernouilli probability of pB = 0.64 with 95% confidence interval [0.47,0.78]. Under the null binomial distribution, B(36, 0.5), the probability of getting at least as many successes as this is p = 0.033. We also evaluated the model for each brain region and found that 30 of the 36 regions (83%) return Bernouilli probability estimates of at least pB = 0.80. Under the null binomial distribution, B(500,0.5), we find that the largest q-value among these 30 regions is q = 4.39 · 10^−54^. Moreover, 24 regions (67%) return Bernouilli probability estimates of at least pB = 0.90 and 8 regions show improvement in all latent spaces, i.e. pB = 1 and q = 0 (Figure 4B). Among these 8 regions are the claustrum, the piriform area, the primary motor and somatosensory areas, and the crus 2. Additional examples of the many regions that demonstrate improvement include: the primary auditory area (pB = 0.83, 95% CI [0.80, 0.86], q = 1.80 · 10^−55^ ), the pallidum (pB = 0.86, 95% CI [0.83, 0.89], q = 3.63 · 10^−65^), and the crus 1 (pB = 0.92, 95% CI [0.90, 0.94], q = 7.68 · 10^−95^). Once again we find that many regions in the sub-cortex do not benefit greatly from the gene expression latent spaces, since the initial gene set was already recapitulating the appropriate match with maximal similarity. We find that the striatum ventral region, caudoputamen, hypothalamus, and pons are maximally similar to their canonical matches in at least 95% of latent spaces. In such cases, the classification approach performs as well as the original approach. While these probability estimates provide a sense of how often an improvement is returned, it is important to note that many regions in this set exhibit a substantial degree of variance over the latent spaces in the ranking of the canonical pairs, e.g. the primary auditory area (μ = 9, 95% CI [1, 19]), the visual areas (μ = 18, 95% CI [7,29]), the paraflocculus (μ = 16, 95% CI [2,29]). This is especially apparent for cerebellar regions, indicating some instability in the neural network<milestone-start />’<milestone-end />s ability to recover these matches.”

We also updated the caption to Figure 4 (370-377).

To quantify the results presented in Figure 5, panels A and B, we used linear regression models. For panel A, we used a simple linear regression to model the average maximal correlation values against a binary variable indicating whether a region was a sensorimotor or supramodal region. For panel B, we used a linear mixed-effects regression to model the average maximal correlation against the type of cortex. A random intercept term was used to model the latent spaces. The results were described in the manuscript (lines 401-416):

“We assessed the similarity between mouse and human isocortical areas using the pairwise correlations in each of the gene expression latent spaces returned from the multi-layer perceptron. For every region in the mouse isocortex, we evaluated the distribution of maximal correlation values over latent spaces (Figure 5A). While the region-wise variance for each isocortical area was large, we found that, on average, sensorimotor regions exhibited higher maximal correlation values than supramodal regions (linear regression with binary predictor: β = −0.042, 95% CI [−0.087,0.003], t(17) = −1.854, p = 0.0812). The mouse primary somatosensory (r = 0.96, 95% CI [0.93, 0.98]) and motor (r = 0.95 with 95% CI [0.92, 0.98]) areas have the highest average maximal correlation values. We additionally examined the distributions of maximal correlation, grouped by cortex type (Figure 5B). To generate these distributions, we computed average maximal correlation values by cortex type in each of the latent spaces. Here too we find that sensorimotor regions are associated with higher maximal correlation values on average compared with supramodal areas (linear mixed-effects regression: β = −0.042, 95% CI [−0.044, −0.040], t(499) = −49.9, p < 2 · 10^−16^). These distributions demonstrate that sensorimotor isocortical regions exhibit more similarity overall on the basis of homologous gene expression than do supramodal regions.”

We also updated the caption to Figure 5 (lines 465-474).

We did not perform any statistical tests for Figure 6, but the body of the text has been updated to reflect the changes induced by the updated human pre-processing pipeline.

We additionally included a new subsection in the “Materials and methods” section to describe these changes. The section is titled “Statistical modelling” (lines 832-861):

“To quantify the improvement in the mouse-human matches when using the latent spaces versus the original gene expression space (Figures 3 and 4), we used a set of logistic regression models to estimate the probability that the rank difference was less than or equal to zero. To estimate the overall improvement due to the latent spaces, we created a binary variable to encode whether the average rank difference over latent spaces for each region met the success criterion. This variable was then used as our target in a logistic regression with no regressors. Once the model was fit, we applied the logistic function to the intercept parameter estimate to get the corresponding estimate for the Bernoulli probability, pB. This transformation was also applied to the bounds on the variance estimate for the intercept to get the corresponding confidence interval. Using the estimated Bernouilli probability, we calculated the corresponding number of successes, k. We then evaluated the probability of obtaining at least k successful outcomes under the null binomial distribution, B(n,0.5). The parameter n was taken to be the number of brain regions under consideration. We additionally applied this approach on a region-wise basis to evaluate the likelihood of a region seeing improvement in the latent spaces. In this case, the null distribution was B(500,0.5) for each region. The resulting p-values were adjusted for multiple comparisons using the false-discovery rate method (Benjamini and Hochberg, 1995). These models were implemented using the glm function from the stats package in the R programming language. In our comparison of sensorimotor and supramodal cortical regions (Figure 5), we used linear models to evaluate the impact of cortex type on maximal correlation values. In the first instance, we computed each region<milestone-start />’<milestone-end />s average maximal correlation over all latent spaces. We then regressed those average values against a binary variable indicating whether the regions were sensorimotor or supramodal. Here we used a simple linear regression. In the second instance, for each latent space we computed average maximal correlation values for sensorimotor regions and supramodal regions. We then regressed these average values against a binary variable as described above. In this case we used a linear mixed-effects regression with a random intercept term to control for observations coming from the same latent space. These models were implemented in the R programming language. The simple linear regression was implemented using the lm function from the stats package, while the linear mixed-effects regression was implemented using the lmer function from the lme4 package. The lmerTest package was used to estimate the degrees of freedom in the mixed-effects model and perform hypothesis testing.”

Reviewer #2 (Recommendations for the authors):I think the manuscript is very polished as-is. I have a number of questions/suggestions that should be considered optional:1) Line 61: "the connections of a brain region tend to be unique". I know exactly what the authors mean (each brain region has a unique/specific connectivity profile), but the sentence could perhaps be clearer.

This sentence has been replaced with the following (lines 67-70):

“It has previously been demonstrated that brain regions can be identified via their unique set of connections to other regions in the brain. This *connectivity fingerprint* can therefore be seen as a diagnostic of an area (Rogier B. Mars et al., 2018a; Passingham et al., 2002).”

2) Why use a multi-layer perceptron to map homologues, as opposed to a more interpretable, SVD-based method, such as PLS or CCA?

We used a classifier rather than a decomposition approach in order to increase the information value found in the transcriptomic data. While the SVD methods are nice in that they jointly decompose both the mouse and human data, the objective functions are such that the resulting variables do not improve the locality of the brain matches. This is especially true if the variable set is truncated after the modelling, e.g. by selecting the first k canonical variables, etc. In our first attempt at implementing the classification approach, we initially opted for a more interpretable class of models, namely multinomial logistic regressions with LASSO regularization. However, these models failed to converge on the 67-label classification task and we soon moved on to more powerful classifiers. We decided to use a neural network approach rather than a tree-based method, because we wanted to be able to extract a set of latent space variables to use for our pairwise correlation analysis.

As you rightly pointed out, the move towards more complicated models reduces the interpretability of the resulting latent spaces. However, in the case of neural networks, we can use feature attribution methods like integrated gradients to extract information about how the resulting classifications or latent spaces relate to the input genes. For more details, please see our response to recommendation #3 from reviewer #1.

Still, the idea of a model that is jointly optimized on the mouse and human data is an attractive one. In the future it may be possible to adapt the classifier approach to use information from both the mouse and human data sets in the construction of the latent space variables.

3) It is still not entirely clear to me how well the perceptron performs in the more conventional, global sense – is there a final, cross-validated accuracy? Is this accuracy significantly greater than what would be expected by chance?

Using our ad hoc resampling cross-validation strategy, the optimal classifier returns an average validation accuracy of 0.597.

To determine the accuracy that we would expect by chance, we ran a data simulation exercise in which we randomly assigned one of the atlas labels to each voxel in our training set. Rather than giving every label equal weight, we estimated the probability of drawing a given label using the proportion of training voxels with that label. For instance, 2955 voxels have the “Caudoputamen” label, and so that probability of drawing that label in our simulation is 2955/51219 = 0.0568, where n = 51219 is the total number of voxels. Thus, we randomly drew 51219 labels and computed the resulting accuracy. We repeated this simulation 10000 times to get a null distribution of accuracy scores. The resulting null accuracy was 0.029 with 95% CI [0.028, 0.031]. This is slightly larger than the expected 1/67 = 0.015 that would result if all labels were given equal weight. So, the neural network’s validation accuracy of 0.597 is much greater than what we would expect by chance.

4) In most of the analyses, there is a clear distinction between the cortex and cerebellum, which should then be expected to drive the configuration of the latent spaces. Have the authors attempted to perform the analysis using cortex only?

Please see our response to recommendation #1 from reviewer #1. In summary, we trained the multi-layer perceptron to classify only the mouse isocortical regions. We found that the resulting latent spaces improve the variance in the ranks of the canonical neuroanatomical homologues for certain isocortical regions, but don’t substantially improve the central tendencies of these rank distributions. We also found that supramodal regions exhibit higher maximal correlation values than sensorimotor regions in these latent spaces. In the clustering analysis, the sensorimotor isocortical regions no longer cluster together when using the isocortical latent spaces rather than the original latent spaces. Interestingly, the scree plot suggests that the optimal clustering solution might have as many as 10 clusters.

We chose not to pursue this approach for the current paper, since the work needed to do it properly would amount to an entirely new research project.

5) Do the authors have a sense of what biological pathways the homologous genes are involved in?

To get a sense of what biological pathways are over-represented by the homologous gene set, we ran a gene enrichment analysis. We obtained a data set of biological process modules from the Bader Lab at the University of Toronto. These modules are lists of genes involved in different biological processes, e.g. “nervous system development”. Then for each of the modules, we ran a hypergeometric test to identify whether our set of homologous genes was over-represented in the module compared with the full gene set, i.e. whether the proportion of homologous genes in the module set was larger than the proportion in the full set. The resulting p-values were corrected for multiple comparisons using the Benjamini-Hochberg method. We found that 938 modules were significant at a q-value threshold of 0.001. These modules were saved to a CSV file, to be included with the manuscript as Supplementary File 1. The 10 most significantly over-represented modules are:

1. Nervous system development

Regulation of multicellular organismal process

Generation of neurons

Regulation of biological quality

Neurogenesis

System development

7. Multicellular organism development

8. Regulation of nervous system development

9. Regulation of nervous system development

10. Regulation of localization

11. Anatomical structure development

These results are described in the Results section titled “Homologous genes capture broad similarities in the mouse and human brains” (lines 120-126):

“Using a gene enrichment analysis, we found that this reduced gene set was significantly associated with a number of biological processes related to the nervous system, with Gene Ontology labels such as <milestone-start />“<milestone-end />nervous system development”, <milestone-start />“<milestone-end />neurogenesis”, and <milestone-start />“<milestone-end />regulation of nervous system development”. Additional modules returned with high significance were <milestone-start />“<milestone-end />regulation of multicellular organismal process”, <milestone-start />“<milestone-end />regulation of biological quality”, and <milestone-start />“<milestone-end />multicellular organism development”. The full set of significant modules can be found in Supplementary File 1.”

We also included a new section in “Materials and methods”, titled “Gene enrichment analysis” (lines 753-764):

“We ran a gene enrichment analysis on the set of homologous genes obtained from the NCBI HomoloGene database. We first downloaded Gene Ontology data for biological process related modules from the Bader Lab at the University of Toronto (http://baderlab.org/GeneSets). These data include a gene set of 16563 genes and a module set of 15757 biological process modules. Every module is associated with a subset of genes from the full gene set. For each module, we used a hypergeometric test to evaluate whether the homologous gene set was over-represented in the module subset, compared with the full gene set. The resulting p-values were adjusted for multiple comparisons using the false-discovery rate method (Benjamini and Hochberg, 1995). A total of 938 modules were found to be significant at a threshold of 0.001. The surviving modules were ordered according to their p-values and written out to a comma-separated values data file (Supplementary File 1). This analysis was carried out using the `tmod` package in the R programming language.”

The full set of enriched modules is made available as a CSV file in Supplementary File 1.